# Telomere-to-telomere *Phragmites australis* reference genome assembly with a B chromosome provides insights into its evolution and polysaccharide biosynthesis

Jipeng Cui [1,2], Rui Wang[1,2], Ruoqing Gu [1,2], Minghui Chen[1], Ziyao Wang[1,2], Li Li[1,2], Jianming Hong[1] & Suxia Cui [1,2] ✉

*Phragmites australis* is a globally distributed grass species (Poaceae) recognized for its vast biomass and exceptional environmental adaptability, making it an ideal model for studying wetland ecosystems and plant stress resilience. However, genomic resources for this species have been limited. In this study, we assembled a chromosome-level reference genome of *P. australis* containing one B chromosome. An explosion of LTR-RTs, centered on the Copia family, occurred during the late Pleistocene, driving the expansion of *P. australis* genome size and subgenomic differentiation. Comparative genomic analysis showed that *P. australis* underwent two whole gene duplication events, was segregated from *Cleistogenes songorica* at 34.6 Mya, and that 41.26% of the gene families underwent expansion. Based on multi-tissue transcriptomic data, we identified structural genes in the biosynthetic pathway of pharmacologically active *Phragmitis rhizoma* polysaccharides with essential roles in rhizome development. This study deepens our understanding of Arundinoideae evolution, genome dynamics, and the genetic basis of key traits, providing essential data and a genetic foundation for wetland restoration, bioenergy development, and plant stress.

*P hragmites australis* are widely distributed in rivers, lakes, dunes, alkaline salt flats, and other habitats worldwide, with strong environmental adaptability and colossal biomass. The high cellulose content of *P. australis* stalks can be used as an excellent raw material for paper production and as pasture for large animals. Moreover, *P. australis'* complex and extensive root network can absorb many kinds of heavy metal ions, and is widely used for water purification, wetland protection and soil stabilisation[1]. It has essential ecological and economic values in ecological protection, animal fodder, and traditional Chinese medicine.

*Phragmitis rhizoma* (fresh or dried rhizome of *P. australis*), which has been used clinically in China for over 2000 years, is known as Lugen in Chinese medicine. Its pharmacological effects have been documented in several ancient medical books, such as the Invaluable Prescriptions for Emergencies (Bei Ji Qian Jin Yao Fang), Yu Qiu's Exegesis for Materia Medica (Yu Qiu Yao Jie), and so on[2]. In modern medical research, Lugen is effective in

antimicrobial[3], anti-inflammation[4,5], antioxidative and hepatoprotective[6], and antiviral[7]. Lugen has been widely used in the novel coronavirus pneumonia (COVID-19) outbreak with significant efficacy[8,9]. The Lugen polysaccharides have been shown to be the significant medicinal constituents of *Phragmitis rhizoma* (PR), in which the acidic polysaccharide PRP-2 has been well characterized in terms of its pharmacological mechanism and structure[3–6]. PRP-2 consists mainly of galactose (34.70%), fucose (36.15%), and rhamnose (0.88%) in the form of →3)-β-D-GalpA-(1 → , →2, 3)-α-L-Fucp-(1→ and α-L-Fucp (4SO3-) -(1 → [4]. However, due to the complexity of genome ploidy and fewer genetic resources of *P. australis*, which seriously limits molecular mechanistic studies and germplasm improvement in *P. australis*.

With the continuous improvement and development of High fidelity reads sequencing (HiFi) and high-throughput chromosome conformation capture (Hi-C) technologies, the resolution of B chromosome (Bs) has been significantly improved[10]. Bs are often thought not to have any function, but

[1]College of Life Sciences, Capital Normal University, Haidian District, Beijing, China. [2]Beijing Key Laboratory of Plant Gene Resources and Biotechnology for Carbon Reduction and Environmental Improvement, Beijing, 100048, China. ✉e-mail: sxcui@cnu.edu.cn

large-scale histological studies are now characterizing the specific functions of Bs in certain species[11,12]. As B chromosomes have been discovered and assembled in more species, their functions have been more deeply analyzed[13–20]. The presence of Bs can affect the phenotype and reproduction of organisms, particularly affecting pollen formation and fertilization in plants. The Bs can preferentially attach to the spindle on the egg side in female meiosis to segregate and produce unequal gametes[13–15]. Bs was observed to have nondisjunction during the first pollen mitosis in *Secale cereal* and *Aegilops speltoides*. Moreover, Bs in maize were observed to undergo nondisjunction in the second meiotic division, and sperm nuclei with two Bs would preferentially fertilize the egg cell[21]. In addition, active genes or non-coding RNAs in Bs can also affect maize A-chromosome (As) gene transcriptional profiles or phenotypes[18–20].

Here, we report an allotetraploid *P. australis* genome at the telomere-to-telomere chromosome level, which was successfully disassembled into two sets of subgenomic haplotypes. Based on FISH and Hi-C results, we unexpectedly discovered and assembled the B chromosome in *P. australis*. With the full-length transcriptome data from multi-tissue hybrid sequencing, we obtained more accurate genome annotation information. Through phylogenetic and evolutionary analyses, we determined that *P. australis* and *Cleistogenes songorica* shared a common ancestor approximately 34.6 million years ago and inferred a chromosomal evolutionary trajectory in *P. australis* that contained two whole genome duplication (WGD) events. Combined with tissue-specific transcriptome WGCNA analysis, we identified a series of genes associated with polysaccharide biosynthesis in Lugen. Our work deepens our understanding of plant evolution in the Arundiaceae. It provides a vital genetic resource for exploring tissue-specific Lugen polysaccharide biosynthesis and investigating the regulation of rhizome development by carbohydrates.

## Results

### *P. australis* genome assembly and annotation

We found a reed strain (*Phragmites australis* subsp. Cuiplus) with an above-ground portion of stem height >350 cm, basal diameter >1.5 cm, and highly developed underground rhizomes (Fig. 1a), with a genome size predicted by flow cytometry to be approximately 1.8 Gb[22]. Karyotyping and fluorescence in situ hybridization (FISH) demonstrated that the *P. australis* specimen exhibited 50 chromosomes, predominantly proximal or mid-stranded chromosomes. The hybridization signals of the 5SrDNA and 18SrDNA repeat sequences probes were characterized by one pair of weak and one pair of strong hybridization signals (Fig. 1a). Genome survey data (48.31 Gb) was employed for K-mer analysis (k = 17) to facilitate a more detailed characterization of the *P. australis* genome. The size of this *P. australis* genome was 1685.39 Mb, the heterozygosity rate was 0.61%, and the proportion of repetitive sequences was 50.40%, making it a complex genome with high assembly difficulty (Supplementary Fig. 1a). Furthermore, the GenomeScope results indicated that the aaab < aabb and the Smudgeplot demonstrated that this *P. australis* genome exhibited a greater propensity towards the AB haplotype genome (Supplementary Fig. 1b). In light of the aforementioned evidence, it was initially hypothesized that this *P. australis* is a heterozygous aneuploid tetraploid prior to the formal assembly of its genome.

We obtained 482.94 Gb subreads data and 32.53 Gb HiFi reads data using PacBio Sequel II platform genome sequencing. Using hifiasm (v 0.16.1) for the primary assembly of the genome, we obtained a *P. australis* genome sketch with a size of 874.62 Mb, containing 312 Contig sequences, N50 of 33.94 Mb, and GC content of 44.21% (Table 1). The genome sketch features were similar to the flow cytometry and genome survey sequencing analysis results. Subsequently, we used Hi-C data (113.26 Gb) to visualize Contig's alignment order and orientation for error correction (Supplementary Table 1). Finally, we anchored 99.46% of the allele sequences to 25 chromosomes with 34 Mb for N50 and 10 for L50 (Table 1). The obtained chromosome-level genomes had a gap on Chr 6B and Chr B, respectively. Subsequently, the gap on the Chr 6B chromosome was successfully filled using TGS-GapCloser software and HiFi data. All chromosomal telomeres (AAACCCT) were successfully assembled, including 19 chromosomes with telomeres assembled at both ends (Supplementary Tables 2 and 3).

Using various genomics data and tools, we identified three candidate *P. australis* centromere repeat sequences in the *P. australis* genome with 108 bp, 216 bp, and 324 bp. Meanwhile, we obtained the genome-wide 5-methylcytosine (5-mC) locus information for *P. australis* from HiFi data using pb-CpG-tools. We found that regions of reduced gene density and increased density of repetitive sequences on *P. australis* chromosomes highly overlapped with areas of dense 5-mC loci (Supplementary Fig. 2). There is growing evidence that the chromosomal centromere region contains fewer genes and denser repetitive sequences and that DNA in this region tends to be highly methylated[23–28]. Finally, we successfully localized the candidate regions of centromere in *P. australis* chromosomes based on the 5-mC distribution (Supplementary Fig. 3).

We combined multiple methods to assess the quality of our assembled *P. australis* genome (PaCui.No1) exhaustively. The genome-wide Hi-C interaction mapping showed that our assembly results conformed to inter- and intra-chromosomal interaction requirements (Supplementary Fig. 4a). The GC content versus sequencing depth distribution map showed that our genome had no significant contaminating sequences during sequencing and assembly (Supplementary Fig. 4b). The CEGMA and BUSCO assessments were 95.97% and 99.30%, respectively (Supplementary Fig. 4c, d). By mapping the survey and iso-seq data to the genome, the mapping rate and coverage exceeded 98%, indicating that our assembled genome has high integrity and uniformity of sequencing (Supplementary Table 4, Supplementary Fig. 4e). Merqury concordance assessment showed that all chromosomes' consensus Quality Value (QV) values exceeded Q40, indicating that the accuracy and confidence of our genome assembly exceeded 99.99% (Supplementary Table 3). The LTR Assembly Index (LAI) reached 11.54, which has reached the reference level genome (Supplementary Fig. 4f). The single nucleotide polymorphism (SNP) and insertion/deletion (InDel) were 0.97% and 0.06%, respectively (Supplementary Table 5), and they were relatively evenly distributed across the chromosomes (Fig. 1h, i). As demonstrated in Table 1, an evaluation of the documented assembly metrics for the *P. australis* genomes indicates a notable enhancement in the quality of the PaCui.No1 assembly[29–31]. Furthermore, synteny analysis with the recently reported *P. australis* T2T genome (CN) demonstrated that, with the exception of a large structural variation (Inversion) between certain chromosomes, such as Chr B, Chr 1B, Chr 2B, Chr3B, and Chr 4B, the two genomes exhibited a high degree of conservation and consistency in the majority of structural regions (Supplementary Fig. 5). These quality assessment metrics indicate that our assembled *P. australis* genome has ultra-high continuity and integrity, providing a genetic basis for subsequent in-depth studies of centromere and highly repetitive regions (Fig. 1b).

We combined homology-based and ab initio prediction to identify repetitive sequences in the *P. australis* genome (Supplementary Fig. 6). 64.98% (553,687,485 bp) of the sequences were annotated as transposable elements (TEs), of which 394,357,364 bp were identified as long terminal Repeat transposons (LTRs), which accounted for 46.28% of the *P. australis* genome (71.22% of TEs); 10.77% of the genome sequences were annotated as DNA transposons (16.58% of TEs); and 3.02% (25,707,169 bp) of the sequences were annotated as Tandem Repeat, wherein, 2,965,766 bp were identified as simple sequence repeats (SSRs) containing si- (56.47%), di- (29.61%), tri- (11.68%), tetra- (1.63%), penta- (0.37%), and hexa- (0.23%) repeat units. We predicted 41,008 protein-coding genes in the *P. australis* genome, with an average gene length and coding DNA sequence (CDS) length of 4758 bp and 1242 bp, respectively. The BUSCO assessment of these genes amounted to 98.40% (Table 1). Gene function annotation of the protein-coding genes annotated within the genome of *P. australis* was performed based on Swiss Prot, KEGG, InterPro, GO, TrEMBL, and NR gene function databases. The combined results showed that 40.527 (98.83%) of the 41,008 protein-coding genes in *P. australis* could be annotated in at least one of the databases (Supplementary Fig. 7). Meanwhile, we identified non-coding RNAs in the *P. australis* genome, including 229 miRNAs, 836 tRNAs, 1744 rRNAs, and 459 snRNAs. We identified three miRNAs (Chr 7B, Chr 7 A, and Chr 5B), two rRNAs (Chr 6B and Chr 4B), and two tRNAs

**Fig. 1 | Genomic characterization of *P. australis*.**
**a** Morphology and Fluorescence in situ hybridization (FISH). The *P. australis* tiller stage, heading stage, Lugen (Rhizomes of *P. australis* after removal of fibrous roots, cleaning, sectioning, and drying), and three-year-old rhizomes were shown separately. The heading stage of *P. australis* was grown at the experimental site of the College of Life Sciences, Capital Normal University. The complete chromosome was identified by telomere repeat sequence probes (green). The 5SrDNA probe (red arrow) showed purple fluorescence, and the 18SrDNA probe (green arrow) showed green fluorescence. Scale bars 5 μm. **b** Circos plot of the *P. australis* genome. a. Chromosome length. b. GC content. c. GC skew. d. Gene density. e. LTR-Gypsy density. f. LTR-Copia density. g. Tandem repeats density. h. SNPs. i. InDels. j - m. miRNA, rRNA, snRNA, and tRNA densities, respectively. The inner lines indicate collinear blocks. Only linkages between homologous chromosomes are shown here. All densities were calculated in a 100,000 bp window.

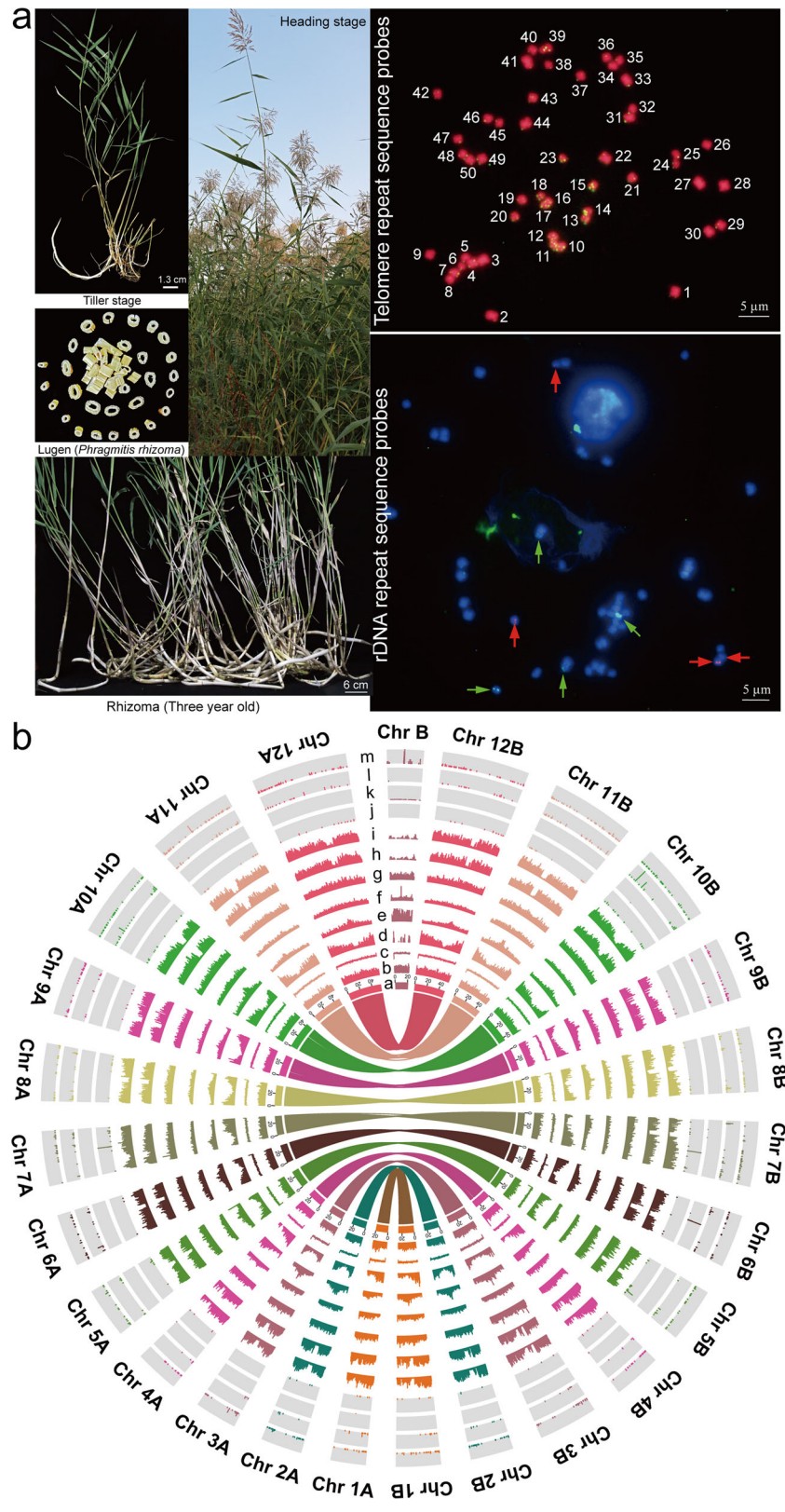

(Chr B and Chr 11 A) densely populated regions in the *P. australis* genome (Fig. 1b, j–m and Supplementary Fig. 8).

### Introns in the *P. australis* genome have driven increased gene lengths

Compared to Poaceae relatives, there was a notable decline in the proportion of genes with lengths below 2000 bp, while the proportion of genes exceeding 5000 bp increased significantly in *P. australis* (Fig. 2a). In contrast, the overall distribution of introns, exons, and CDS lengths of *P. australis* was not significantly different from that of the related species. However, the proportion of genes containing introns was significantly higher at 89.91% (Fig. 2b, Supplementary Table 6). Subsequently, a more comprehensive structural characterization of the 11,654 genes in *P. australis* with a length exceeding 5,000 bp was conducted (Fig. 2c). The results demonstrated that

**Table 1 | *P. australis* genome assembly statistics**

| Contig | PaCui.No1 | Draft genome[30] | LpPhrAust1.1[31] | CN (Chinese lineage)[29] |
|---|---|---|---|---|
| Total size (bp) | 874,619,212 | 1,139,927,050 | - | - |
| Number of contigs | 312 | 13,411 | - | - |
| Number of contigs ≥ 50,000 bp | 152 | 4617 | - | - |
| Largest contig (bp) | 54,964,116 | 3,219,705 | - | - |
| GC content (%) | 44.21 | 44.04 | - | - |
| N50 length (bp) | 33,936,801 | 194,574 | - | - |
| L50 count | 11 | 1370 | - | - |
| BUSCO (%) | 99.3 | 93.3 | - | - |
| Chromosomes | | | | |
| Number of chromosomes+Unchr | 25 + 50 | - | 24 + 13 | 25 + 1288 |
| Total size (bp) | 852,113,342 | - | 849,281,002 | 920,401,352 |
| GC content (%) | 44.27 | - | 44.17 | 43.94 |
| N50 length (bp) | 34,052,747 | - | 35,123,032 | 34,219,654 |
| Gaps | 1 (Chr B) | | 59 | 0 |
| L50 | 10 | - | 10 | 11 |
| BUSCO genome (%) | 99.3 | - | 98.9 | 98.9 |
| CEGMA (%) | 99.46 | - | - | |
| LTR Assembly Index (LAI) | 11.54 | - | - | 12.71 |
| Second-generation Data Mapping | 98.62% | - | - | - |
| Third-generation Data Mapping | 99.99% | - | - | - |
| Annotation | | | | |
| Number of genes loci | 41,008 | 64,857 | 47,513 | 42,498 |
| BUSCO annotation (%) | 98.40 | - | - | 98.2 |
| Functional annotation (%) | 98.83 | - | - | - |
| Percentage of repeat sequences (%) | 65.62 | 56.19% | - | 63.95% |
| Percentage of TEs (%) | 64.98 | - | - | - |
| Number of predicted SSRs | 160,796 | - | - | - |
| Number of rRNAs | 1744 | - | - | 3749 |
| Number of tRNAs | 836 | - | - | 851 |
| Number of miRNAs | 229 | - | - | - |
| Number of snRNAs | 459 | - | - | - |
| Subgenome | Number of chromosomes | Length (bp) | Number of genes | LTR Assembly Index (LAI) |
| Subgenome A | 12 | 399249732 | 20290 | 12.24 |
| Subgenome B | 12 | 427051217 | 20425 | 12.47 |
| B Chromosome | 1 | 21198758 | 277 | 8.31 |

the genes with more than ten introns constituted 29.46% (3433) of these genes. The lengths of introns and exons exhibited a concentration in the range of 3000 bp to 5000 bp and 1 bp to 3000 bp, respectively, which were markedly higher than their respective mean values (mean length of introns: 720.66 bp, mean length of exons: 301.18) (Supplementary Table 6). A strong positive correlation was observed between intron length and gene length for these genes (Pearson R = 0.99, $p < 2.2e-16$) (Fig. 2d, e). It is noteworthy that 2473 genes with ultra-long introns (introns exceeding 10 kilobases in length) were identified in *P. australis*. The ultra-long introns did not appear to exert a deleterious effect on gene expression. In addition, certain genes exhibited significantly elevated tissue-specific expression (Fig. 2f). The utilization of full-length transcriptome data from the PacBio platform for assisted annotation and optimization of gene structures has resulted in a notable enhancement in the accuracy and comprehensiveness of the gene annotations. Consequently, it is unlikely that these gene structures with ultra-long introns result from annotation errors. The accuracy of the structural annotation of these genes was confirmed by mapping the RNA-seq and Iso-seq data to representative genes with tissue-specific expression (with ultra-

long introns) (Supplementary Fig. 9). Furthermore, the introns of these genes contained a substantial number of transposon sequences (Supplementary Fig. 9), which may have contributed to the formation of these ultra-long introns, thereby driving the increase in gene length.

## Subgenome sorting and B chromosome assembly
Due to the lack of genetic information on the genomes of *P. australis* parents, subgenome sorting is highly difficult. By intra-species syntenic map constructed from 13,788 paralogous homologous gene pairs in *P. australis*, we found 12 pairs of very strong one-to-one syntenic blocks between 25 chromosomes (Fig. 3a). Combining the *Oryza sativa* and *Panicum virgatum* genomes, we extracted homologous chromosome pairs with *P. australis*, respectively, and used their single-copy genes to calculate genetic distances (Fig. 3b, Supplementary Fig. 10). We classified the chromosome with closer affinity to the homologous chromosome of *O. sativa* into subgenome A (LAI: 12.24), while the other one was categorized into subgenome B (LAI: 12.47). We successfully sorted out two haplotype subgenomes that reached the reference genome level (Ordered by chromosome length from largest to

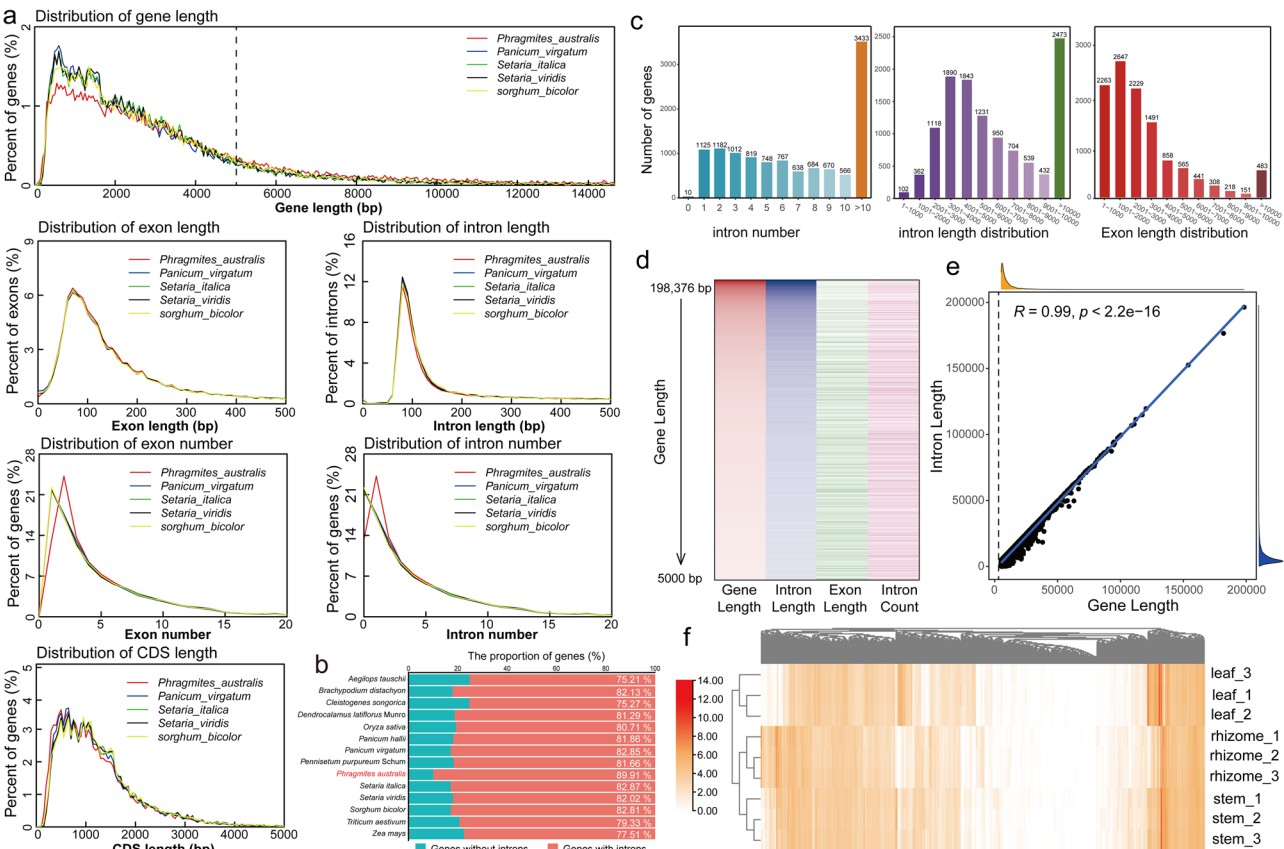

**Fig. 2 | Characterisation of gene structure in *P. australis*. a** Comparison of five Poaceae gene elements. The analysis encompasses the distribution of gene length, the distribution of exon and intron length and number, and the distribution of coding sequence length. **b** Statistics on the number of genes with introns in 14 Poaceae species. **c** The following section presents a statistical analysis of the structural characterization of genes with a length greater than 5000 bp in *P. australis*. The number and distribution of intron lengths, the number of introns, and the lengths of exons were

enumerated separately. **d** Heatmap of the characteristics (length and number) of genes with length greater than 5000 bp. **e** Correlation between gene length and total intron length. The correlation between total intron length and gene length was measured using the Pearson correlation coefficient for these genes with lengths greater than 5000 bp. **f** The heatmap illustrates the expression of genes with ultra-long introns. The data were expressed as fragments per kilobase of transcript per million mapped reads (FPKM) and underwent logarithmic transformation prior to visualization.

smallest) (Supplementary Table 3). The subgenomes showed an overall 1:1 covariance but frequent chromosomal structural variations between some chromosomes, such as chromosomal inversions between Chr 8B and Chr 10 A, Chr 12 A and Chr 12B, and chromosomal translocations between Chr 10 A and Chr 7B, and Chr 8 A and Chr 10B (Supplementary Fig. 11).

We found no significant colinear blocks of chromosome 25 within *P. australis* or between related species (Fig. 3a, Supplementary Fig. 10). This distinctive co-linearity outcome was similarly documented in the maize genome B chromosome investigation, which may represent a distinctive attribute of the B chromosome[32]. The quality and integrity of the Chr25 chromosome assembly were verified by sequencing depth and coverage of short-read sequencing data (Supplementary Figs. 4e and 12a). Furthermore, the continuity of the assembly of a high-density region (10 MB in length) of tRNA on this chromosome was confirmed by PCR experiments (Supplementary Fig. 12b and Supplementary Table 7). It has a significantly reduced density and number of protein-coding genes, with only 277 genes present (only 9.65% of the length of this chromosome). Homologs of the 268 genes in Chr25 are widely dispersed in the 24 normal chromosomes of *P. australis* (E value ≤ 1e-5) (Fig. 3c). In addition, 87.88% (18,602,795 bp) of the sequences in chromosome Chr25 were identified as transposon sequences, and 9.96% (2,111,980 bp) of the sequences were TRFs. Evolutionary analysis of the complete LTR-RT sequences showed a close relationship between the transposon sequences of Chr25 and those of chromosome A (Supplementary Fig. 13). We found many rRNA sites, multiple significant tRNA density peaks widely distributed in this chromosome, and the presence of multiple possible centromere regions (Fig. 1b and S3). To investigate whether these

retained genes have certain specific functions, we performed GO enrichment of genes on Chr B (Supplementary Fig. 14). We found that these genes are mainly involved in biological processes related to sister chromatid movement in meiosis, pollen sperm cell differentiation, and spindle. In the Molecular Function classification, these gene products are mainly involved in "histone H3-methyl-lysine-4 demethylase activity", "histone demethylase activity", "histone H3-methyl-lysine-4 demethylase activity", and "telomeric DNA binding". The 82 genes on chromosome 25 were expressed to varying degrees in different tissues, with a mean FPKM value greater than 1. The GO enrichment analysis revealed that these expressed genes are involved in germ cell production ("sexual reproduction", "pollen sperm cell differentiation", "cellular process involved in reproduction in a multicellular organism", "gamete generation", "fertilization forming a zygote and endosperm", and "fusion of sperm to egg plasma membrane involved in double", etc.), cell membrane and transmembrane transport (plasma membrane fusion, oxidoreduction-driven active transmembrane transporter activity), and energy metabolism metabolism ("NADH dehydrogenase activity", "electron transfer activity", "oxidoreductase activity, acting on NAD(P)H, quinone or similar compound as acceptor", etc.) related functions (Fig. 3c, d). In conclusion, a B chromosome was constructed that was notably shorter than the 24 A chromosomes, and this Bs may play an active role in the reproductive biology of *P. australis*.

## Transposons have driven genome expansion and evolution

Many long terminal repeat-retrotransposons (LTR-RTs) exist in plant and animal genomes, especially in plant genomes. In *P. australis*, the

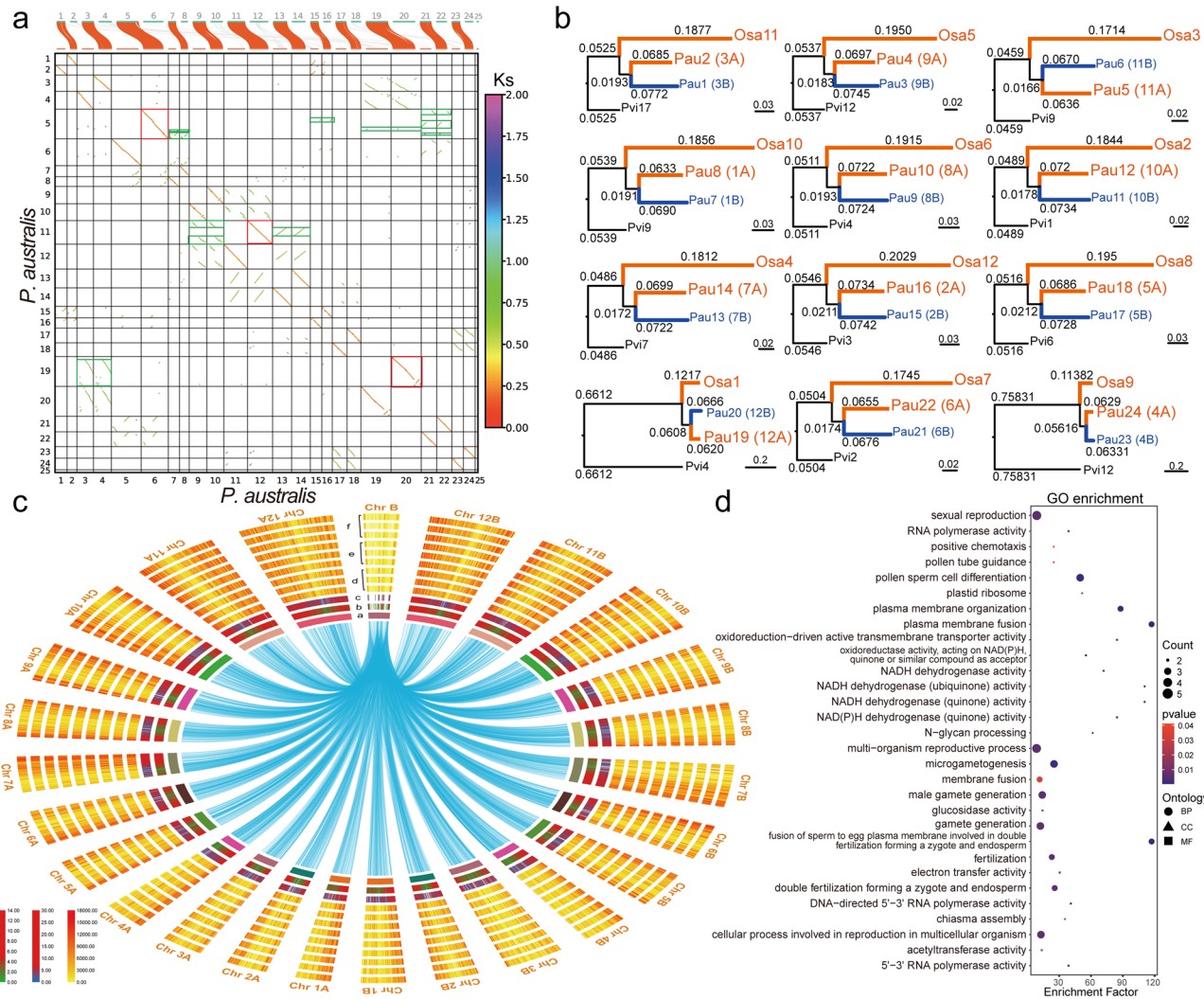

**Fig. 3 | Subgenomic isolation and B chromosome identification. a** Dotplot of co-orthologs genes and Ks within the *P. australis* genome. The color of the dot indicates the Ks of the gene pair. **b** Phylogenetic tree of genetic distances constructed based on single-copy homologous sequences. **c** Distribution of genes in the *P. australis* B chromosome in normal chromosomes. a Chromosomes. b Gene density. c Distribution of the coverage of multi-tissue mixed full-length transcriptome sequencing data. d - f. Distribution of leaf, aerial stem, and rhizome RNAseq data coverage, respectively. The blue connecting line in the figure indicates homology between the genes and those in the B chromosome with an E-value < 1e-10. **d** GO enrichment of genes expressed in the *P. australis* B chromosome. Genes with FPKM mean > 1 in chromosome B were retained for GO enrichment using clusterProfiler 2.0.

transposons were mainly LTRs, of which the Gypsy and Copia families accounted for 34.73% and 21.65% of the total LTR length, respectively (Supplementary Fig. 6). The distribution density of the Gypsy family on *P. australis* chromosomes was strongly negatively correlated with the gene arrangement, and it was mainly distributed near the centromere. Still, the Copia family was relatively uniformly distributed at both ends of *P. australis* chromosomes, which was consistent with the trend of gene distribution (Fig. 1b, Supplementary Fig. 2). By calculating the timing of LTR-RT insertions, we found that, except for the *C. songorica* genome, the genomes of *P. australis* and the other gramineous species experienced LTR-RT insertion events of varying magnitude between 0.10 and 0.16 Mya and all of them had higher average chromosome lengths than *C. songorica* (Fig. 4a). The amplification densities and peak insertion times of LTR-RTs in these genomes differed considerably, and both *P. australis* genomes underwent at least two significant transposon rapid insertion events (Fig. 4b, Supplementary Fig. 15).

As observed in *Dracaena cochinchinensis*[33] and *Leymus chinensis*[34], two transposon insertion events of different magnitudes were identified in the *P. australis* genome. *P. australis* experienced an insertion event of LTR-RTs centered on Gypsy families between one and two million years ago. These

LTR-RTs are defined as ancient LTR-RTs (Fig. 4b). We found that these ancient Gypsy families were concentrated near the centromere of certain chromosomes, e.g., Chr 9 A, Chr11B, Chr8B, Chr2A, etc., which not only increased the length of repetitive sequences within the centromere region but also drove centromere evolution (Supplementary Fig. 16a). Subsequently, a more intensive and rapid insertion event of LTR-RTs occurred 0-500,000 years ago, in which the Copia family was significantly dominant, and we define the LTR-RTs from this event as new LTR-RTs (Fig. 4b). These newly inserted Copia family members accounted for 62.05% of all Copia and were widely distributed in or near gene regions (Supplementary Fig. 16b). We conducted evolutionary analyses of Copia and Gypsy sequences extracted during the two insertion events, respectively. The results demonstrated that a notable expansion of the Gypsy family occurred during the ancient insertion event. This result is consistent with the evidence that the Gypsy family experienced two periods of significant expansion, as illustrated in Fig. 4c, d. The large number of insertions of these elements may affect gene sequences in the vicinity of TEs to a certain extent, providing abundant raw material for variation in genome evolution. Interestingly, we found that the content of Copia family sequences in these genomes conformed to a linear normal distribution between the genome size and the

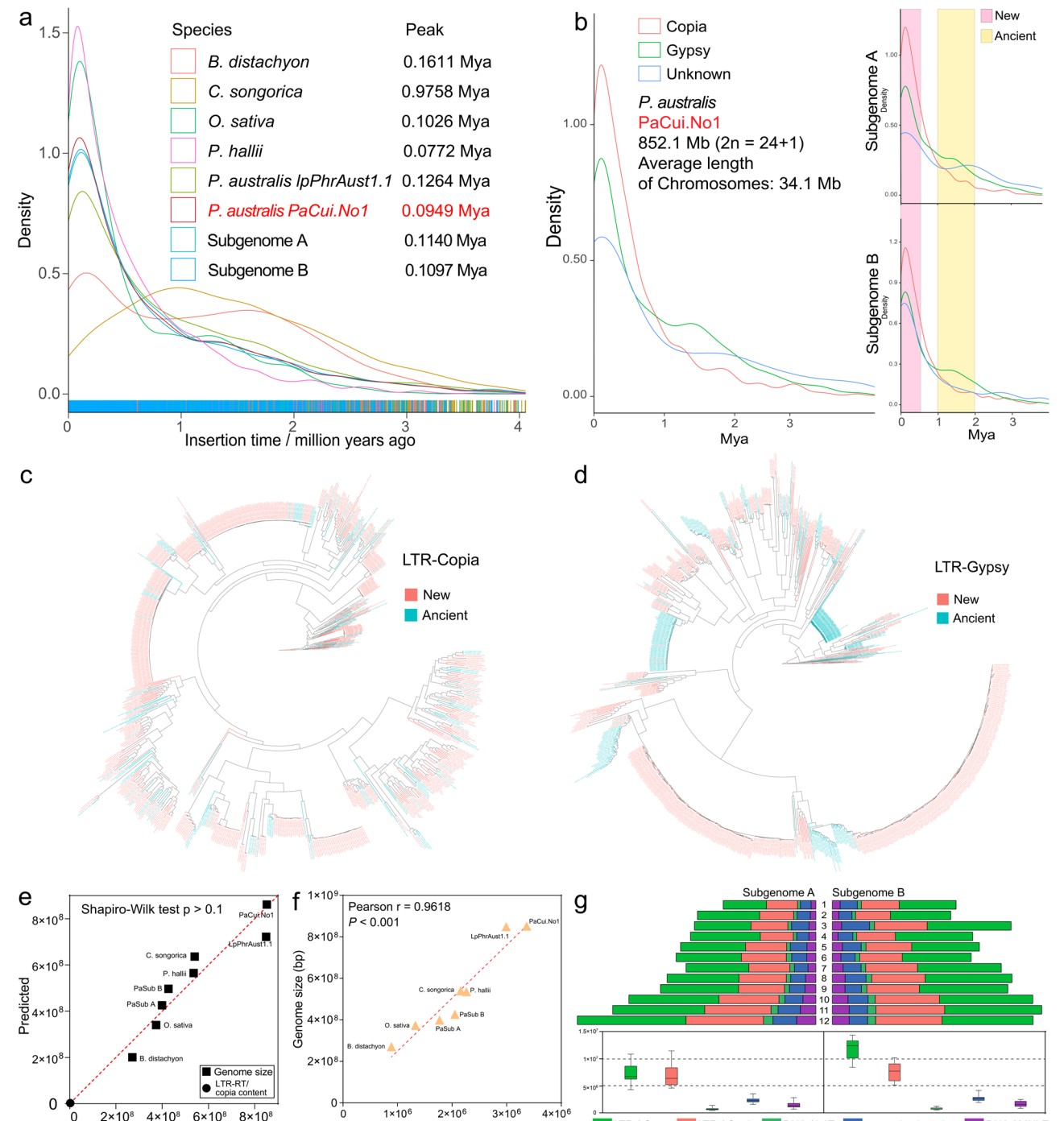

**Fig. 4 | Analysis of LTR-RTs in *P. australis* genomes. a** Insertion times of LTR-RTs in eight genomes, including two *P. australis* subgenomes. **b** Insertion times of significant types of *P. australis* LTR-RTs. The right panel indicates the significant types of LTR-RT insertion times for subgenome A and subgenome B, respectively. Pink and yellow shading indicate new and old transposon insertion events, respectively. **c**, **d** Evolutionary trees constructed from transposon sequences generated in new and old transposon insertion events in the Copia (**c**) and Gypsy (**d**) families. Evolutionary trees were constructed based on the complete reverse transcriptase structural domain sequences in the LTR-RTs sequences and visualised using ggtree (R package). The LTR-RT sequences are distinguished by different colours in the phylogenetic tree; further details can be found in the treefile (https://doi.org/10.6084/m9.figshare.27016279). **e** Normal QQ plot of Copia family content versus genome size for the *P. australis* genome. The normal distribution test using the Shapiro-Wilk (SW) method. **f** Pearson correlation analysis between Copia family content and genome size in the *P. australis* genome. **g** Differences in transposon content between two sets of subgenes in *P. australis*.

Copia family sequences (Fig. 4e), and there was a very strong positive correlation (Pearson R = 0.9618, *p* < 0.001) (Fig. 4f).

The LTR-RT insertion events and extant transposon content occurring between the two sets of subgenomes in *P. australis* exhibited notable discrepancies (Fig. 4b, Supplementary Fig. 16c–f). To gain further insight into the role of transposons in the subgenomes, we extracted the complete reverse transcriptase domain sequences for subfamily identification and evolutionary analysis. The Copia and Gypsy family types in subgenomes A and B were essentially the same. A comparison of subgenomes A and B revealed that subgenome B contains two Chlamyvir members and no Alesia members (Supplementary Fig. 16c, d). Copia and Gypsy are two separate superfamilies with nine major branches and six major branches, respectively

(Supplementary Fig. 16e, f). The number of Gypsy elements in subgenome B (1,155) is greater than that in subgenome A (1,020). Furthermore, the total length of Gypsy subfamily members is also significantly greater than that identified in subgenome A (Fig. 4g, Supplementary Fig. 16c, d). These results suggest that gypsy insertion events in the *P. australis* genome tend to be more frequent in the B chromosome. The Gypsy family insertion event led to differences in transposon types and lengths in the two subgenomes and to some extent, drove the expansion and differentiation of the subgenome B.

## Gene families, phylogenetic relationships, and WGD

To understand the patterns of gene family divergence during *P. australis* evolution, we analyzed the *P. australis* genome for gene family clustering with 13 Gramineae species, including *Aegilops tauschii*, *Brachypodium distachyon*, *Cleistogenes songorica*, *Dendrocalamus latiflorus* Munro, *Oryza sativa*, *Panicum hallii*, *Panicum virgatum*, *Pennisetum purpureum* Schum, *Setaria italica*, *Setaria viridis*, *Sorghum bicolor*, *Triticum aestivum*, *Zea mays*, and *Arabidopsis thaliana* as outgroup species. We identified 28,025 gene families in 15 species and 17,494 gene families in *P. australis*, with multiple-copy ortholog genes accounting for 32.21% of all gene families (Fig. 5a). All species shared 5507 gene families, and 297 unique gene families were present in *P. australis*, containing 871 genes (Supplementary Fig. 17a). We found that these *P. australis* unique family members are mainly involved in biological processes related to protein or macromolecule depalmitoylation, "protein dephosphorylation", "lipoprotein catabolic process" and "phosphatidic acid metabolic process". The molecular functions were also enriched for the terms "palmitoyl hydrolase activity", "phosphoprotein phosphatase activity", "protein serine/threonine phosphatase activity" and "phospholipase A2 activity". Moreover, many genes related to vesicular transport, such as Golgi and vesicles, were enriched in the Cellular Component category. In addition, some genes were also enriched in Terms related to oleoresin lactone synthesis and reactive oxygen species biosynthetic process. The GO functional enrichment of these *P. australis* unique genes seems to predict the existence of special competence in depalmitoylation and dephosphorylation with other species (Supplementary Fig. 17b).

Surprisingly, expansion occurred in 41.26% of gene families in *P. australis*, involving 7217 gene families (20,142 transcripts), a much higher proportion than in the other 14 species. The GO functional enrichment results indicated that these expanded gene families were primarily associated with the functions of telomere maintenance (GO:0032200, GO:0010833, GO:0007004, and GO:0000723), DNA binding (GO:1990837, GO:0043565, GO:0000976, and GO:0003690), and starch synthesis (GO:0004556 and GO:0016160) (Supplementary Fig. 17c). Gene family expansion in *P. australis* has resulted in a significantly increased proportion of Multiple copy orthologs, while providing *P. australis* with more members of genes with potential functions that can help *P. australis* to undergo rapid adaptation and evolution in the face of unfavorable environments and stresses. Meanwhile, we found 288 positively selected genes in 1,039 single-copy direct homologs in *P. australis*. Many of these genes are present in genes involved in telomere maintenance (GO:0032204, GO:0000723, and GO:0032200) as well as DNA damage checking (GO:0007095, GO:0031572) (Supplementary Fig. 17d). These genes subjected to positive selection related to telomere or DNA repair play an essential role in the perennial mechanism of *P. australis*.

We have constructed a phylogenetic tree to investigate the evolutionary origins and species-related relationships of *P. australis* using 302 single-copy genes identified in 15 species (Fig. 5a). The results showed that Arundiaceae is more closely related to Chloridoideae. Their common ancestor appeared earlier than the ancestor of Panicoideae and later than the ancestor of Oryzoideae, and separated from the ancestor of Panicoideae 38.1 million years ago (Fig. 5a). The synteny analysis showed strong colinearity between *P. australis* chromosomes and *O. sativa* and *C. songorica* chromosomes (Fig. 5b). Meanwhile, we found 246 syntenic blocks containing 19,384 collinear gene pairs between *P. australis* and *O. sativa*, and 614 syntenic blocks containing 25,732 collinear gene pairs with *C. songorica*. Colinear genes with C. songorica accounted for 62.75% of all genes in *P. australis*,

much higher than the collinear gene pairs with *O. sativa* (Supplementary Table 8). These results suggest that *P. australis* and *C. songorica* are evolutionarily closely related sister groups that diverged about 34.6 million years ago. Colinear blocks of some chromosomes in *P. australis* mapped to more than two chromosomes in *O. sativa* and *C. songorica* simultaneously (Fig. 5b).

We found in the genome dot plot that each chromosome of *P. australis*, except Bs, had one best-matched chromosome and two sub-matched chromosomes where more chromosomal rearrangement events occurred, suggesting that two WGD events occurred in *P. australis* (Fig. 3a). The depth of colinearity between the *P. australis* genome and *O. sativa* (1: 2) and *C. songorica* (2: 2), respectively, predicted that *P. australis* experienced a shared WGD event with *O. sativa* and an independent WGD event with *C. songorica* (Supplementary Fig. 18, Supplementary Table 8). Two significant peaks in the *P. australis* genome, corresponding to the two WGD events, were analyzed by synonymous substitution (Fig. 5c). Five Poaceae genomes, including two subgenomes, share an ancient ρ-WGD event between ~60 and 80 Mya. Following this, *P. australis* was successively separated from *O. sativa* (Ks = 0.470) and *C. songorica* (Ks = 0.305) and underwent a polyploidy event unique to *P. australis* at ~23.9 Mya (Ks = 0.211) (Fig. 5c). The four-fold degenerate sites (4DTv) (Supplementary Fig. 19) analysis was consistent with the Ks results. Meanwhile, we observed significant peaks at both Ks = 0.628 and Ks = 0.205 in the comparison between subgenes A and B, corresponding to the ancient ρ-WGD event and the polyploidy event unique to the *P. australis* genome, respectively. In summary, we hypothesize that the parental ancestor of this *P. australis* diverged from successive *O. sativa* and *C. songorica* after undergoing the ρ-WGD event shared by most gramineous species and formed an allotetraploid *P. australis* through interspecific hybridization and an exclusive heterologous tetraploidization event (WGD-IV). We used WGDI based on the Ancestral monocot karyotype except for Acoraceae (AMK-A) to construct a karyotype evolutionary process that includes the major species of Gramineae. Accompanied by two WGD events, the eight chromosomes of AMK-A underwent at least 356 chromosome breaks, 340 chromosome fusions, and eventual integration into the 24 haplotype chromosomes now found in *P. australis* (Fig. 5d). Subgenome A, more closely related to *O. sativa*, experienced a higher frequency of ancestral chromosome splits and fusions and has more collinear gene pairs with other species. These results suggest that subgenome A appears to be the more ancient parental genome. (Fig. 5d, Supplementary Table 8).

## Tissue-specific transcriptome analysis and identification of Lugen polysaccharide biosynthetic genes

In order to gain insight into the biosynthetic pathway of polysaccharides in *P. australis* and the transcription factors (TFs) that play a pivotal role in regulating this process, we conducted a weighted correlation network analysis (WGCNA) on the transcriptome data of three tissue samples of *P. australis*, comprising leaves, aerial stems, and rhizomes (Fig. 6). Principal component analysis revealed high homogeneity among similar tissue transcriptome data and showed significant differences overall (Fig. 6a). The genes with an average FPKM value greater than one were divided into 14 co-expression modules comprising 22,285 genes (softpower = 12). Additionally, three modules (MEturquoise, MEbrown, and MEblue) were identified as being highly correlated with the distinctive tissues of *P. australis*, with a Pearson correlation coefficient exceeding 0.95 (Fig. 6b). The GS-MM results demonstrated that these genes exhibited a high degree of correlation with tissue specificity and played a pivotal role within the module, as illustrated in Fig. 6c. The genes in MEturquoise, which are highly associated with leaves, are mainly involved in photosynthetic pigment synthesis, the electron transport chain, and biological processes related to light-responsiveness. The MEbrown associated with aerial stem tissues contained genes of sucrose or monosaccharide transporter protein families, and the expression of these genes was increased explicitly in aerial stems (Fig. 6d–f). Interestingly, the MEblue module significantly associated with underground rhizomes of *P. australis* was enriched with a large number of genes related to plant

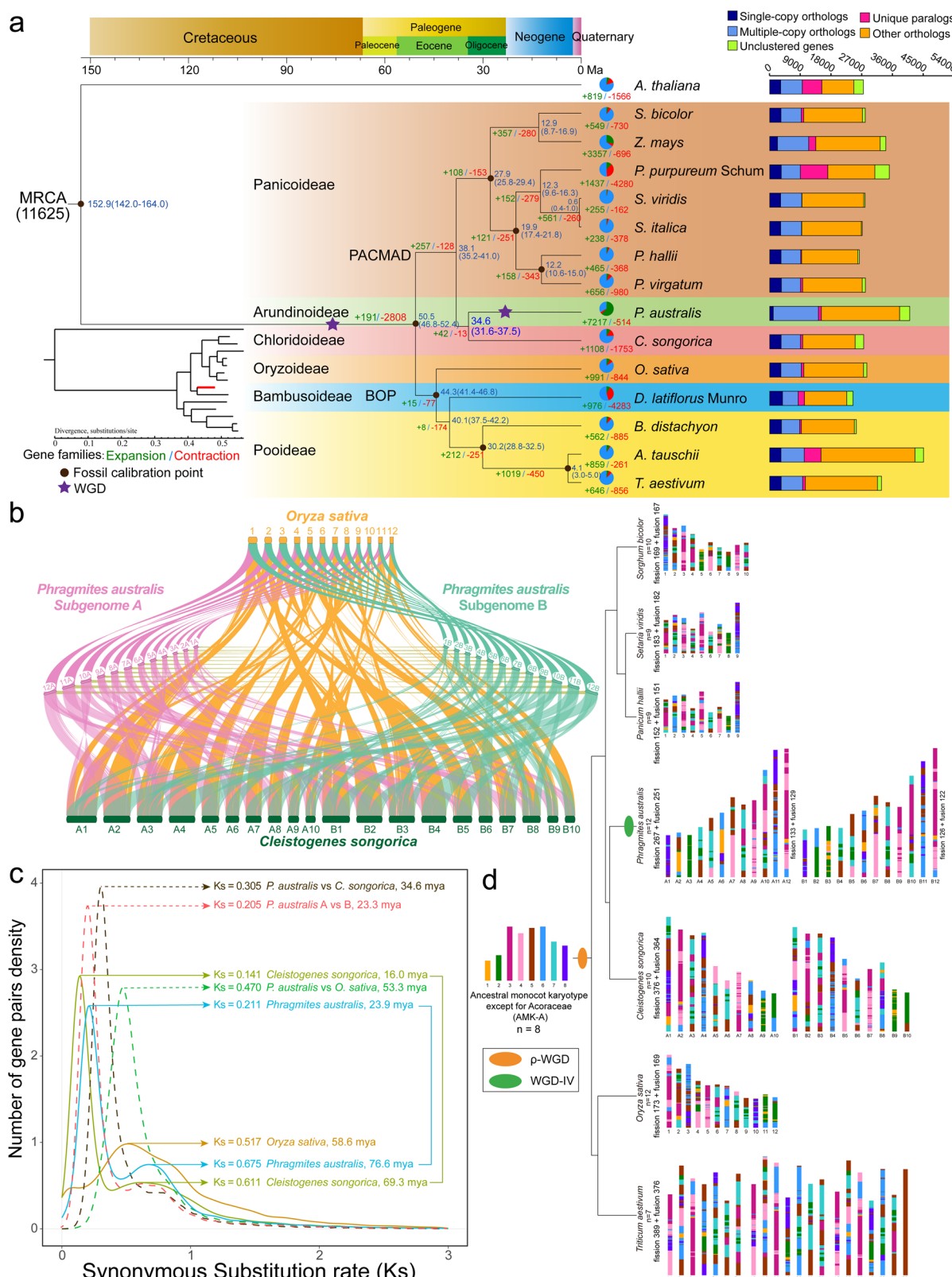

polysaccharide synthesis and cell wall formation, which play essential roles in polysaccharide synthesis and rhizome development of *P. australis*.

Here, we extracted and determined the polysaccharide content of different tissues of *P. australis* using the 'ethanol subsiding method'. The findings revealed that among the diverse tissues of *P. australis*, the rhizomes exhibited the highest polysaccharide content, reaching 25.65 mg/g (Table 2).

Notwithstanding the observation of a greater quantity of crude polysaccharide precipitation in the leaves, the polysaccharide content was found to be exceedingly low (2.04 mg/g.WD), manifesting as a brown powder (Fig. 6g). This phenomenon may be attributed to the fact that a multitude of intricate physiological processes (e.g., photosynthesis, respiration, metabolic activities, etc.) occurring in the leaves result in the production of a

**Fig. 5 | Evolutionary and comparative genomic analysis. a** Phylogenetic analysis, divergence time estimation, and gene family expansion/contraction analysis. A phylogenetic tree was constructed with *Arabidopsis thaliana* as an outgroup using the genomes of 14 Poaceae, including *P. australis*. Purple stars indicate genome-wide replication events, The branch length of a phylogenetic tree represents the amount of cumulative evolution or cumulative mutation; The blue numbers indicate divergence times; and the red and green numbers indicate gene families for expanded and contracted gene families, respectively; Classification of orthologous and lineage-specific gene families in *P. australis* and other representative plants are shown on the right. **b** Collinear relationship of two subgenomes of *P. australis*, *Oryza sativa*, and *Cleistogenes songorica*. Different color lines connect matched gene pairs between different genomes. **c** Distribution of Ks between *P. australis* and the other two species. The lines indicate the distribution of Ks within genomes (continuous) and between genomes (dashed lines). **d** Ancestral karyotype evolution in *Sorghum bicolor*, *Setaria viridis*, *Panicum hallii*, *Phragmites australis*, *Cleistogenes songorica*, *Oryza sativa*, and *Triticum aestivum*. Different colors represent chromosome segments of the Ancestral monocot karyotype except for Acoraceae (AMK-A), and different color combinations represent the chromosomal recombination events that the modern karyotype of each species underwent.

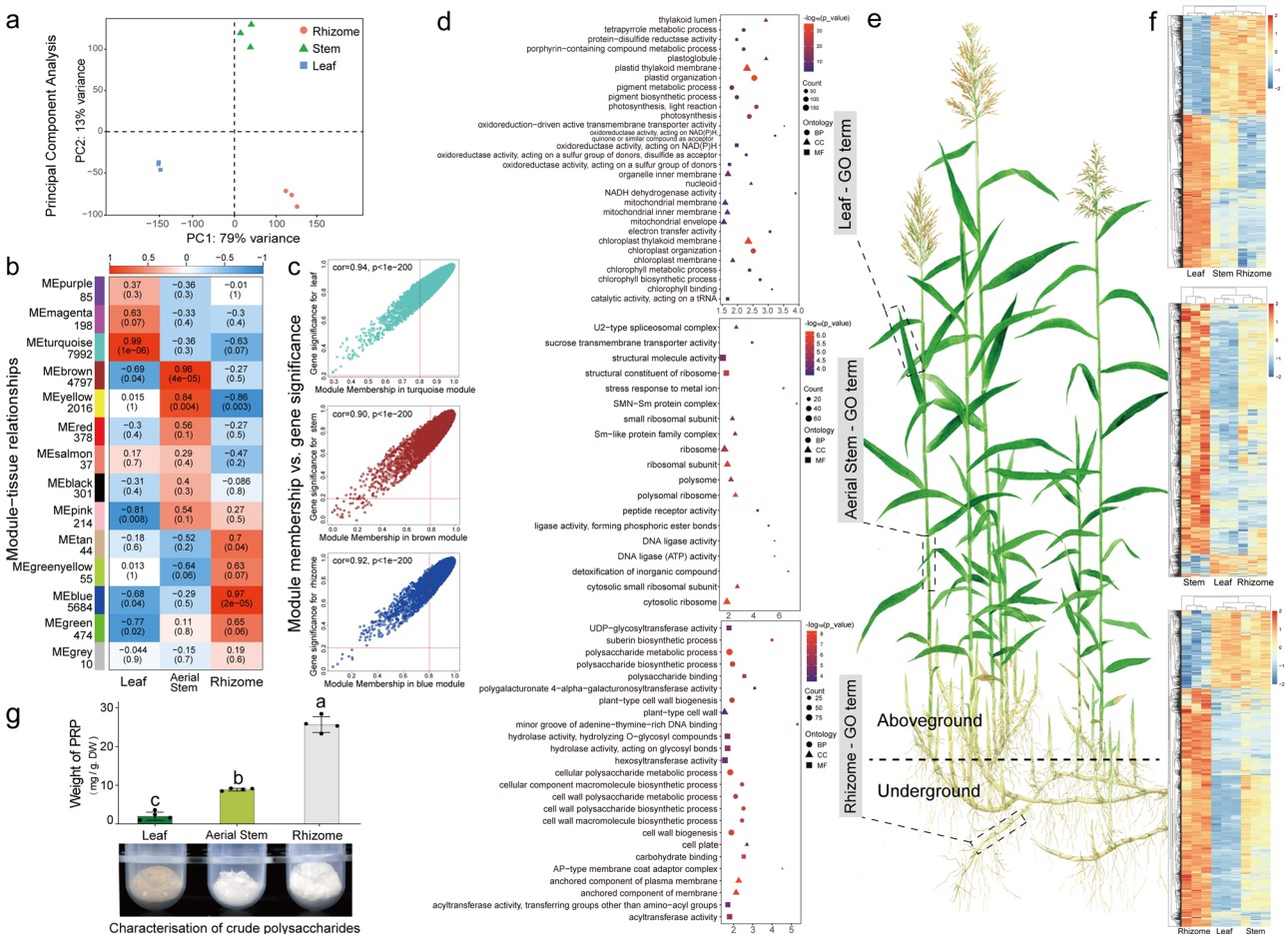

**Fig. 6 | Construction of the *P. australis* co-expression network. a** Score scatter plots for the PCA model with nine samples. **b** Heatmap of gene co-expression network module-tissue association. Each row corresponds to a module characteristic gene (eigengene), and each column corresponds to a specific tissue. Each cell contains the corresponding correlation value and P-value, and indicates the strength of the correlation according to the color. **c** A scatterplot of Gene Significance (GS) for weight vs. Module Membership (MM) in three module eigengene. There is a highly significant correlation between GS and MM in this module, illustrating that genes highly significantly associated with a trait are often also the most critical elements of modules associated with the trait. The red lines indicate the thresholds for |MM| > 0.8 and |GS| > 0.2, respectively. **d** Bubble diagram showing the results of GO enrichment of genes contained in the module eigengene. **e** Pattern map of *P. australis* (drawn by Congying Li). **f** Heatmap showing changes in module eigengene expression (FPKM) after row-scaling treatment. **g** *Phragmites rhizoma* Polysaccharide content in leaves, aerial stems, and rhizomes of *P. australis*. Polysaccharide mass (mg) = crude polysaccharide mass (mg) × polysaccharide concentration (%). Significance derived using a t-test, and different letters indicate significant differences ($P < 0.05$), $n = 4$.

considerable quantity of substances, including polyphenols and pigments (Fig. 6d-f). These metabolites were co-precipitated with polysaccharides when treated with 80% ethanol and oxidized to a brown color during extraction and lyophilization.

There was also a correlation between certain modules and organisations (|Pearson cor| > 0.80). A significant negative correlation (Pearson cor = −0.81) with leaf blades was observed for genes in the MEpink module mainly involved with microtubules or the cytoskeleton (Supplementary Fig. 20a). The down-regulation of the expression of these genes not only reduced the mechanical stiffness of the cell wall but also the maintenance of

cellular tension and expansion pressure, resulting in a softer leaf morphology. *P. australis* is typically a densely tufted tall graminoid, and the softer leaves facilitate the capture of scattered and low-angle light in the presence of external forces such as wind and gravity. The MEyellow module demonstrated a significant positive correlation with aerial stems (Pearson cor = 0.84), as well as a significant negative correlation with rhizomes (Pearson cor = −0.86) (Supplementary Fig. 20b). The genes in MEyellow are primarily associated with ribosome biogenesis, rRNA processing, protein synthesis, and protein folding-related functions. In comparison to the rhizome, which is a nutritive organ, the aerial stems required large amounts

**Table 2 | *Phragmites rhizoma* Polysaccharide (PRP) and protein content in different tissues of *P. australis***

| Tissue | Weight of crude PRP (mg/g.WD) (*n* = 3) | Weight of PRP (mg/g.WD) (*n* = 4) | Weight of Protein (mg/g.WD) (*n* = 4) |
|---|---|---|---|
| Leaf | 51.87 ± 9.41[a] | 2.04 ± 1.03[c] | 0.70 ± 0.07[a] |
| Aerial Stem | 17.91 ± 1.66[c] | 8.90 ± 0.32[b] | 0.17 ± 0.03[b] |
| Rhizome | 37.59 ± 2.43[b] | 25.65 ± 2.05[a] | 0.10 ± 0.08[b] |

Note: total sugar standard curve is $y = 2.2414x + 0.0197$, $R^2 = 0.9984$; protein standard curve is $y = 1.2706x + 0.0114$, $R^2 = 0.9935$. Different lowercase letters (a, b, c) indicate significant differences among groups ($p < 0.05$), determined by one-way ANOVA and LSD post hoc comparisons. Groups sharing the same letter are not significantly different. Data are presented as mean ± SD.

of proteins to carry out functions such as water and nutrient transport, response to environmental stimuli and signalling. The rapid growth of *P. australis* places high demands on the correct folding and assembly of newly produced proteins in the aerial stems. The high level of gene expression in the MEyellow module ensures fast, accurate and efficient protein synthesis to meet the needs of rapid growth.

Based on our functional annotation information, we identified structural genes in the Lugen polysaccharide (PRP2) biosynthesis pathway that encompasses the sucrose synthesis (SUS), sucrose transporters protein (SUC), and Lugen polysaccharide monomer (GDP-Fucose, UDP-Rhamnose, UDP-Galactose and UDP-Galacturonate) pathways, involving a total of 182 transcripts (69 transcripts with average FPKM expression >1). As illustrated in Fig. 7a, the majority of these gene families involved in PRP synthesis have experienced a degree of expansion, as indicated by the red font. The significantly high expression of Sucrose-6F-phosphate phosphohydrolase (SPP) and sucrose phosphate synthase (SPS), two critical enzymes for sucrose synthesis, in leaves increased the amount of sucrose produced via photosynthesis. Sucrose transporter proteins (SUC) are not only related to the mobility and availability of sucrose in plants but also essential for plant-specific tissue development regulation. We identified nine *PaSUC* genes involving 12 transcripts in *P. australis*, among which *PaSUC1.1*, *PaSUC1.2*, and *PaSUC4.1* were significantly overexpressed in aerial stem tissues. They may assume the function of a long-distance sucrose transporter for transporting sucrose synthesized in leaves down through aerial stems to rhizomes. Sucrose synthase (SUS), as a key rate-limiting enzyme in the sucrose synthesis pathway, mediates the reversible conversion of sucrose and ADP (or UDP) to fructose and ADPG (or UDPG). *SUS* (rna-Pau33356.1) expression was significantly increased in *P. australis* rhizomes, which promoted the accumulation of carbohydrates such as sucrose, fructose, and UDP-glucose in the rhizomes. Significantly increased expression of GDP-mannose pyrophosphorylase (GMPP), GDP-mannose 4,6-dehydratase (GMD), and dTDP-4-dehydrorhamnose reductase (UER1) in the polysaccharide monomer synthesis pathway in aerial stems or rhizomes also led to the accumulation of GDP-Fucose and UDP-Rhamnose monomers in these tissues. In addition to the high expression of specific genes that may be associated with galactose stress response in leaves, most of the UDP-glucose-4-epimerase (UGE) and UDP-glucose-4-epimerase (GALE) were significantly overexpressed in rhizomes, and these genes led to the accumulation of UDP-Galactose and UDP-Galacturonate in rhizomes. The high expression of UDP-D-galactose dehydrogenase (UGD) and UTP-glucose-1-phosphate uridylyltransferase (UGP2) in rhizomes may also play an essential role in polysaccharide synthesis in Lugen.

We analyzed cis-acting elements within the 2 kb region upstream (candidate promoter regions) of these *SUC* genes identified in *P. australis* to predict transcription factors regulating sucrose transporter proteins. The promoter regions of these *SUC* genes were found to contain multiple cis-acting regulatory elements associated with phytohormone (abscisic acid and MeJA) or stress response stress (defense and stress, light, and low temperature). Moreover, the binding sites of the transcription factor MYB were also identified, including MYB binding site involved in drought-inducibility,

MYB binding site involved in light responsiveness, and MYBHv1 binding site. The findings indicate that stress-related MYB transcription factors may be a pivotal regulator of sucrose transporter proteins in *P. australis* (Fig. 7b). Since the *PaSUC* genes with an average FPKM > 1 in *P. australis* were all categorized in the brown module, we inferred that the genes in this module were associated with sucrose transport in stems.

In order to identify potential transcription factors regulating the expression of *SUC* genes in the MEbrown module, a screening of the top 100 hub genes was conducted using the Radiality plugin in CytoHubba. Genes with weights lower than 0.40 were filtered in order to ensure the accuracy of the results. Furthermore, the transcription factors in the hub gene co-expression network were identified based on the Plant TFDB (v5.0) database. These Hub genes are mainly involved in sucrose transport and response to heat (Supplementary Fig. 21) and contain nine transcription factors and a sucrose transporter protein (PaSUC1.1) (Fig. 7d). Among the transcription factors co-expressed with *PaSUC*, the expression trend of the *PaMYB* (rna-Pau22479.1) gene in different tissues was found to be highly similar to that of *PaSUC1.1* expression (Fig. 7c). The *PaMYB* was identified by gene family as a member of the R2R3-MYB subfamily. Furthermore, a 6 bp MBS element (CAACTG) was identified within the *PaSUC1.1* candidate promoter region (Supplementary Table 9). The binding probability of the MBS element to the PaR2R3-MYB was then predicted using AlphaFold3 (ipTM = 0.89, pTM = 0.40) (Supplementary Fig. 22). To determine the effect of PaR2R3-MYB on the transcriptional regulation of *PaSUC1.1*, we performed dual luciferase reporter gene (dual-LUC) experiments in *Nicotiana benthamiana* using the PaR2R3-MYB amino acid coding sequence and a sequence 2000 bp upstream of the *PaSUC1.1* gene (Fig. 7e and Supplementary Fig. 23). The pGreenII-62-SK-PaR2R3-MYB (35S::PaMYB) expression vector was employed as the effector, while the pGreenII-0800-LUC-*p*PaSUC1.1 (*p*Pa-SUC1.1::LUC) expression vector, ligated to the *PaSUC1.1* candidate promoter, served as the reporter. In vivo imaging results showed that *N. benthamiana* tissues injected with 35S::PaMYB + *p*PaSUC1.1::LUC displayed stronger chemiluminescent signals compared to empty vector. Further dual luciferase activity assays showed that the relative activity of firefly luciferase (LUC/REN) was significantly elevated in *N. benthamiana* tissues injected with the combination of 35S::PaMYB + *p*PaSUC1.1::LUC (Fig. 7e). In conclusion, a transcription factor (PaR2R3-MYB) was identified that can regulate the high expression of the sucrose transporter protein (PaSUC1.1) in the aerial stem. This provides a foundation for further research into the mechanism of long-distance sucrose transport in *P. australis*.

## Discussion
### Assembly and annotation of the *P. australis* T2T genome
The rapid development of genome sequencing and assembly technologies is facilitating the assembly and understanding of an increasing number of complex genomes, particularly those of medicinal plants[35–37]. This has important implications for the identification of biosynthetic pathways for active compounds and genetic studies of adversity responses in these species. The Arundiaceae is widely distributed worldwide and has enormous biomass and economic value, but only a few genomic resources have been published. In this study, we assembled the high-quality chromosome-scale heterozygous tetraploid *P. australis* reference genome that was successfully phased into two sets of haplotype subgenomes (Fig. 1). The complexity of heterochromatin regions and repetitive sequences has resulted in centromere and telomere sequences being the most challenging regions in genome assembly[38–40]. The identification of telomeric and centromere sequences and candidate regions in the *P. australis* genome was successfully achieved through the association of multi-omics data, thereby providing a genetic basis for the subsequent in-depth study of centromere and highly repetitive regions (Supplementary Fig. 3). Concurrently, a multitude of strategies were employed for the annotation of structural genes, repetitive sequences, non-coding RNAs, and gene functions within the *P. australis* genome (Table 1, Supplementary Fig. 6). A considerable number of genes with lengths exceeding 5000 bp and ultra-long introns were identified in the *P. australis* genome (Fig. 2), which could be attributed to the substantial

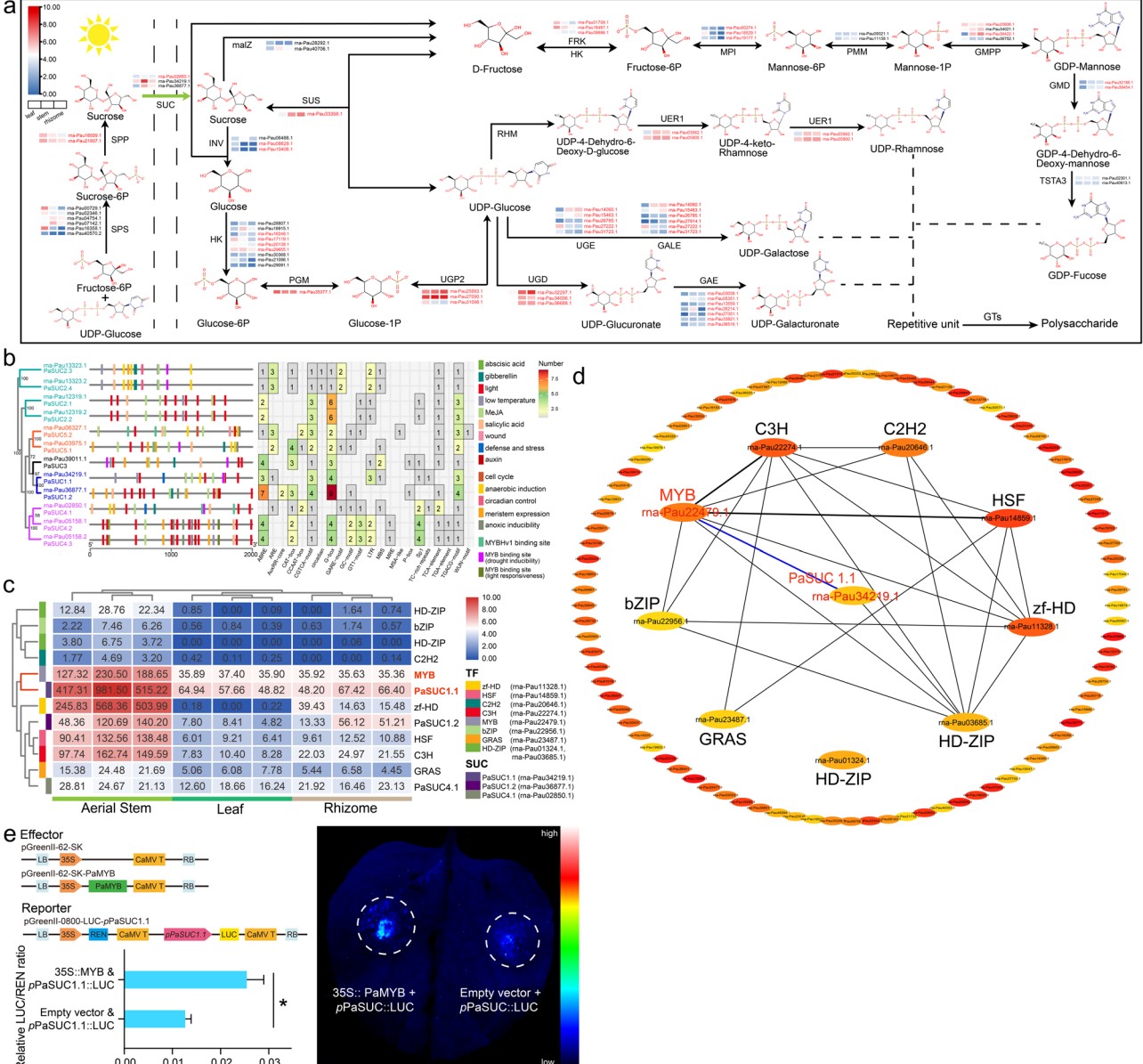

**Fig. 7 | Lugen polysaccharide biosynthesis pathway and sugar transporter protein transcription factor identification.** a Gene expression analysis of genes involved in polysaccharide biosynthesis in "Lugen". Different color blocks represent the expression levels of the encoded genes in different tissues, the squares from left to right correspond to leaves, aerial stems, and rhizomes, and the red font represents the gene family expansion in *P. australis*. Compound structure diagrams were obtained from EMBL-EBI (https://www.ebi.ac.uk/). **b** Prediction of cis-acting elements in the promoter region of the 12 identified *PaSUC* genes. A maximum likelihood phylogenetic tree (left) of 12 PaSUC family members was constructed using IQ-TREE (v

1.6.12) with 1,000 bootstrap replications. **c** Heatmap of sucrose transporters protein expression with transcription factors in Hub gene. The expression level of each gene is represented by log2 (FPKM), with redder colors indicating higher expression levels. **d** Hub genes co-expression network in the brown module. The connecting line indicates the co-expression relationship between TFs and *PaSUC1.1* in hub genes. **e** Dual-luciferase assays demonstrate that PaR2R3-MYB transcriptionally activates *PaSUC1.1*. The data are presented as mean ± SD (*n* = 3), with error bars representing the SDs. Significant differences from the control were determined by Student's t-test (*P* < 0.05).

number of transposon sequences present within the introns. The assembly of this high-quality *P. australis* genome will provide invaluable genetic resources for investigating the evolution of the Arundiaceae, the biosynthesis of medicinal components derived from *P. australis*, and the study of stress response mechanisms.

**Inference of B chromosome function and formation**
The number of samples and genetic studies on plant B chromosomes is currently reported to be low, due to the delayed commencement of research on plant B chromosomes, the inherent difficulties associated with their assembly, and the inherent challenges of genetic transformation in plants[10–12]. In this study, we assembled a B chromosome in *P. australis* with a

length of 21,198,758 bp containing 277 genes (Table 1). The allotetraploid *P. australis* karyotype was ultimately corroborated as AABB (2n = 4x = 48 + 2 Bs) through fluorescence in situ hybridisation. In some haploid reproducing species, e.g., *Nasonia vitripennis*, the Bs called PSR (paternal sex ratio) can eliminate the paternal genome during the mitotic division of the embryo, resulting in the conversion of fertilized female embryos into males. This ensures "selfish" transmission of the PSR chromosome and can lead to sex bias in the entire *N. vitripennis* population[14–16]. In light of the GO enrichment results, we postulate that Bs in *P. australis* may exert an influence or participate in the process of germ cell division, thereby maintaining the presence and propagation of the B chromosome itself in the zygote. A significant inversion in structural variation is observed between our

assembled *P. australis* B chromosome and the recently reported B chromosome (Chinese lineage, CN) (Supplementary Fig. 5). However, the gene clusters that they have retained in evolution demonstrate high conservation in terms of functions relevant to the maintenance of chromosome stability and meiosis[29].This coexistence of functional conservatism and structural variation will provide crucial insights for comprehensive investigations into the evolutionary implications of genomic structural variation, environmental adaptations, and genetic variations among *P. australis* strains.

Here, we provide a detailed characterisation of the assembled B chromosome: (1) Similar to Bs in maize[32], most of the genes in Bs of *P. australis* are highly homologous to those in the A chromosome. However, larger blocks of colinearity could not be detected (Fig. 3). (2) 87.88% of the sequences in the Bs are transposable sequences. Furthermore, the evolutionary analysis of LTR-RTs indicates that these transposable elements are closely related to those in the A chromosome (Supplementary Fig. 13). (3) Multiple candidate centromere regions identified in Bs, which predicts that Bs has undergone multiple breakage, fusion and chromosomal structural rearrangement events (Supplementary Fig. 3). (4) The high density of repetitive and transposable sequences within the chromosomal regions of Bs may have played a role in mediating chromosome breakage and fusion events (Fig. 1b and Supplementary Fig. 3). (5) Bs of *P. australis* serve to prevent genomic instability by maintaining the heterochromatin state and transposon silencing through the establishment of a considerable number of methylation sites (5-mC) (Supplementary Fig. 3). Based on the aforementioned results, we postulate a possible mode of Bs production in *P. australis*: the breakage of the centromere region of the A chromosome, which results in fragmented chromosomes that undergo fusion and recombination to form a multimeric B chromosome. This process is mediated by centromere repeats and transposable sequences, or telomeric repeats, and is thought to play a significant evolutionary role. Subsequently, Bs retained genes favourable for functional maintenance and structural stability through gene rearrangement and selection during long-term evolution. Concurrently, Bs preserved the stable configuration of the chromosome centromere and the transmission of genetic information through extensive methylation modifications. The assembly and study of the *P. australis* B chromosome will provide a valuable genetic resource for elucidating the intricacies and diverse array of ecological adaptations of the *P. australis* genome.

However, the substantial number of repetitive sequences present in the *P. australis* B chromosome represents a significant challenge for complete assembly. In particular, the long segmental repetitive regions, including transposons and tandem repetitive sequences, contribute to the difficulty of gap-filling. Currently, B chromosome studies in maize employ flow-sorting techniques to distinguish B chromosomes from other chromosomes. Additionally, Oxford Nanopore data are used to generate more complete assemblies of B chromosomes. As the assembly of the B chromosome is still in its preliminary stages, there are limited reference sequences available for comparison. As more *P. australis* genomes or B chromosomes are resequenced and completely assembled in the future, they will provide distinctive models for the study of chromosome genetics and chromosome evolutionary mechanisms.

## LTR-RT insertion events have driven *P. australis* genome evolution and expansion

It is well known that frequent climatic oscillations and glacial movements during the late Pleistocene (0.129-0.0117 Mya) brought dramatic changes to the distribution and genetic structure of plants and animals on a global scale and drove speciation and the creation of new species[41–43]. Meanwhile, TE outbursts can bring more evolutionary raw materials for species, which is one of the important factors driving species evolution and new species divergence[44]. LTR-RTs can move through the genome and insert into new sites by a "copy and paste" transposition mechanism[45]. TEs with a large

number of repetitive sequences can lead to gene inactivation, translocation, pseudogenes, and even chromosomal rearrangements due to their "active insertion" characteristics[46,47]. However, the genome size expansion driven by transposons counteracts the genome size reduction caused by the deletion of mutated genes to a certain extent. It maintains the stability of gene size[48]. The *P. australis* LTR-RTs were mainly composed of the Gypsy family, enriched in the *P. australis* centromere region, and the Copia family, which is consistent with the trend of gene distribution (Fig. 1b, Supplementary Fig. 6). Further analysis revealed a strong positive correlation between the Copia family sequence content in the genome and genome size, and the LTR-RTs insertion events were species-specific (Fig. 4 and Supplementary Fig. 15). This suggests that LTR-RT insertion events are widespread in Poaceae and strongly suggests that the Copia family plays a vital role in driving the rapid expansion of genome size and evolution of species. Two distinct LTR-RT insertion events of varying magnitude and type in *P. australis* during the Pleistocene period were responsible for driving genome size expansion and subgenomic differentiation in *P. australis* on the one hand, and for providing the raw material for a large number of gene mutations on the other, which enabled rapid adaptation to the environment. In addition, the uneliminated transposon sequences retained in the ultra-long intron drive the length of *P. australis* genes to some extent (Supplementary Fig. 9).

To reduce the impact of frequent transposon "jumps" on the genome, plants can not only directly inhibit the activation and mobilization of TE activity through DNA methylation[49–53] but also indirectly regulate heterochromatin ratios through histone methylation[54,55]. It was observed that the 5-methylcytosine site in *P. australis* was significantly enriched in the region around the centromere (Supplementary Fig. 3). Therefore, we hypothesize that after two widespread, high-density TE insertion events, LTR-RTs may have been modified by DNA methylation (5-mC) of transposons concentrated in the centromere region. This prevented the expression and "jumping" of which transposons had not yet been eliminated, thus maintaining the stability of the centromeric region of the genome.

## Gene family expansion and WGD events

As we all know, telomere length and stability are essential in the biological life cycle. As the most difficult-to-repair, highly repetitive region in the genome, the existence and length maintenance of the telomere sequence are related to genome stability and cellular lifespan[56–59]. In *P. australis*, 41.26% of the gene families expanded during long-term evolution, mainly involving gene families related to telomere maintenance and lengthening or DNA repair, and these genes were subjected to positive selection simultaneously (Supplementary Fig. 17). This provided for telomere integrity and stability and played an essential role in the evolution of *P. australis*' extreme environmental adaptability and perennial characteristics. Comparative genomic and phylogenetic analyses showed that *P. australis* of the Arundiaceae subfamily is sister to *C. songorica* of the Chloridoideae, consistent with the same previous study[30]. The *P. australis* genome undergoes two WGD events, a ρ event shared with most gramineous species and a heterologous quadruplication event (WGD-IV) that occurs at 23.9 Mya exclusive to *P. australis* (Fig. 5c). Large-scale chromosome fission and fusion events in the *P. australis* genome have resulted in large segments of chromosome alterations and increases in chromosome number, while contributing to some extent to the evolution of the *P. australis* genome (Fig. 5d).

## Tissue-specific WGCNA analysis and identification of biosynthetic pathways of *phragmitis rhizoma* polysaccharides

The rapid asexual reproduction of *P. australis* utilizing an extensive rhizome network can provide an absolute advantage in interspecific competition and is a crucial organ for *P. australis* to achieve perenniality in complex habitats[60,61]. At the same time, the *phragmitis rhizoma* (known as "Lugen" in Chinese medicine) is a medicinal plant used clinically for more than 2000 years, and Lugen polysaccharides are its main medicinally active molecule[3–6]. Most of the sugars produced by photosynthesis in *P. australis* leaves are transported downwards through the aerial stems in the form of sucrose to the rhizome, where they accumulate in forms such as

polysaccharides or starch. However, in the aerial parts (leaves and stems) most of these sugars are either consumed by metabolic pathways or converted into structural polysaccharides (e.g. cellulose, hemicellulose, pectin, etc.), resulting in relatively low polysaccharide content in leaves and aerial stems (Fig. 6g). Differences in the accumulation of polysaccharides in different organs of *P. australis* are determined by a combination of different physiological functions of the organs, sugar metabolic pathways and plant adaptive strategies. Here, we constructed the Lugen polysaccharide (PRP-2) biosynthesis pathway involving the 182 transcripts (Fig. 7a).

By multi-tissue RNAseq analysis, we found that *P. australis* increased the expression of genes related to sucrose synthesis in the leaves and of sucrose transporter proteins responsible for the long-distance downward transport of sucrose in the aerial stems. Sugar content fluctuations resulting from the transport of plant carbohydrates over long distances to the roots can act as signal-regulating hormones or other signaling fluxes to induce the activation of apical meristematic tissues (axillary buds) and promote the development and extension of lateral and branching roots[62–68]. In addition, the adequate sucrose supply and increased sucrose availability in axillary buds can reduce competition for sucrose from the terminal buds and help rhizome axillary buds break through the inhibition of apical dominance[69–73]. Furthermore, in *Oryza longistaminata*, Fan et al. found that the hydrolyzed monosaccharides of sucrose could affect rhizome elongation and orientation by increasing the osmotic pressure of rhizome cells; meanwhile, the sucrose could promote the development of axillary buds into secondary rhizomes and retard the upward growth of axillary buds in rhizomes[74–76]. The high expression of these genes promoted the accumulation of sucrose and polysaccharides in rhizomes and reduced the inhibition of rhizome axillary buds, facilitating the development of the complex rhizome network of *P. australis*. In addition, the high expression of genes in the monomer synthesis pathway of Lugen polysaccharides increased the accumulation of raw materials for cell wall synthesis, laying a rich energy and material foundation for the rapid differentiation and development of perennial rhizomes in *P. australis*. A PaR2R3-MYB transcription factor that can regulate sucrose transporter protein was identified based on WGCNA analysis (Fig. 7c, d). The ability of PaR2R3-MYB to bind to the *PaSUC1.1* promoter and promote *PaSUC1.1* transcription was further confirmed by a dual luciferase assay. In conclusion, the *P. australis* regulates sucrose transporter proteins by up-regulating the expression of PaR2R3-MYB in aerial stems, which continuously translocates sucrose from leaves to rhizomes over long distances and then promotes rhizome development. This regulatory mechanism not only provides a material basis for the accumulation of polysaccharides in rhizomes, but also helps *P. australis* to use rhizomes to rapidly occupy ecological niches in a variety of harsh environments and interspecific competition.

## Materials and methods
### Plant material
In this study, young leaf tissues of biennial *Phragmites australis* (Cav.) var. Cuiplus was collected for genome survey sequencing, HiFi sequencing, and Hi-C sequencing from a sample plot planted with reeds at Capital Normal University. At the same time, different tissues are used for full-length transcriptome sequencing (Nine types of tissues including flower, stem apical meristem, aerial stems, leaves, aerial stem buds, rhizome internodal tissues, rhizome nodal meristem, rhizome buds, and fibrous roots) and transcriptome sequencing (including mature leaves, aerial stems, and rhizome tissues, in three biological replicates) were collected.

### FISH
We first subjected root tip meristematic tissues to laughing gas treatment and obtained dispersed mid-stage chromosome cells using glacial acetic acid fixation and enzymatic digestion with a mixed enzyme solution (cellulase and pectinase 3:1). Subsequently, DAPI staining was used to obtain more explicit chromosome images and accurate numbers. Finally, the samples were subjected to fluorescence in situ hybridization based on a fluorescent probe for telomere-conserved repeats, 5SrDNA, and 18SrDNA universal probes, and photographed under an Olympus BX70 fluorescence microscope.

### DNA extraction
High-quality genomic DNA was extracted from young leaves of *Phragmites australis* using a modified CTAB method. First, we added 5 ml of 2% CTAB, 50 μL of β-mercaptoethanol, and 26 μL of proteinase K to *P. australis* leaf tissues after sufficient grinding (with the addition of PVP), mixed thoroughly, and incubated at 65 °C for 30 min. After centrifugation at 12,000 rpm for 10 min, the supernatant was retained. Next, an equal volume of phenol/chloroform/isoamyl alcohol (25:24:1) was added and mixed thoroughly. After centrifugation at 12,000 rpm for 10 min (repeated 2 times), and retain the supernatant. Continue by adding an equal volume of chloroform/isoamyl alcohol (24:1) extracted once and centrifuged at 120,000 rpm for 10 min and retained the supernatant. Subsequently, precooled isopropanol (4 °C) was added, mixed upside down and allowed to stand for 30 min at 4 °C. The precipitate was centrifuged at 12000 rpm for 10 min and retained. After washing twice by adding 1 ml of pre-cooled 70% ethanol at 4 °C, the precipitate was dried at 37 °C. Finally, the precipitate was dissolved with Tris-HCl and RNaseA and incubated at 37 °C for 30 min, then stored at −20 °C. The quality and quantity of the extracted DNA were examined using a NanoDrop 2000 spectrophotometer (NanoDrop Technologies, Wilmington, DE, USA), Qubit dsDNA HS Assay Kit on a Qubit 3.0 Fluorometer (Life Technologies, Carlsbad, CA, USA) and electrophoresis on a 0.8% agarose gel, respectively.

### Genome sequencing and assembly
This was followed by BGI short reads, PacBio subreads, and Hi-C interactive reads, which were sequenced. First, we performed paired-end reads with an insert size of 150 bp sequencing using the BGI T7 sequencing platform and obtained 47.74 Gb clean reads data after filtering and cleaning by SOAPnuke[77] software. The basic genomic information such as genome size, heterozygosity, the proportion of repetitive sequences, and other genomic information was obtained by K-mer frequency distribution analysis with k = 17 by jellyfish[78] and genomescope software (https://github.com/tbenavi1/genomescope.2.0). Subsequently, the SMRTbell library was constructed using the SMRTbell Express Template Prep kit 2.0 (Pacific Biosciences). Pacbio High-fidelity (HiFi) reads were performed using SMRT Cell on the Sequel II System with Sequel II Sequencing Kit by Frasergen Bioinformatics Co., Ltd. (Wuhan, China). Finally, the Hi-C library was constructed according to previous studies[79]. The methodology is as follows: First, samples were cross-linked under vacuum infiltration for 30 min with 3% formaldehyde at 4 °C and quenched with 0.375 M final concentration glycine for 5 min. The cross-linked samples were subsequently lysed. Endogenous nuclease was inactivated with 0.3% SDS, then chromatin DNA were digested by 100U MboI (NEB), and marked with biotin-14-dCTP (Invitrogen) and then ligated by 50U T4 DNA ligase (NEB). After reversing cross-links, the ligated DNA was extracted through QIAamp DNA Mini Kit (Qiagen). Purified DNA was sheared to 300- to 500-bp fragments and were further blunt-end repaired, A-tailed and adaptor- added, followed by purification through biotin-streptavidin-mediated pull-down and PCR amplification. The quality of the library was ensured by using the Q-PCR method. The Hi-C libraries were quantified and sequenced on the BGISEQ-500 platform (BGI, China). After dejointing the raw sequencing data using Trimmomatic software and filtering the low-quality Reads, we ended up with 113.26 GB of clean data for genome-assisted assembly.

The consensus reads they were generated using ccs software (https://github.com/pacificbiosciences/unanimity) with the parameter '-minPasses 3'. After quality control, we obtained 32.53 Gbps of HiFi reads data. These long (~15 kb) and highly accurate (>99%) HiFi reads were assembled using hifiasm v0.16.1[80] with the parameter '-l3', yielding a total of 312 contigs containing the initial assembled genome. Then, we applied 3D-DNA to order and orient the clustered contigs. The Juicer[81] was used to filter and cluster the sequences, and the Juicebox was applied to adjust chromosome construction manually. We formed a Hi-C-assisted pre-assembly genome sketch sequence

containing 79 Contigs after heterozygous and redundant sequence filtering based on Hi-C interaction signals and NT comparison results. Next, by examining allelic interactions in the Hi-C heatmap, the order and orientation of alleles on the pseudochromosomes were evaluated and adjusted. Finally, we anchored 79 contigs to 25 chromosomes, and the effective scaffolding rate of Contig was 99.46%. The HiFi data can provide highly accurate sequences from long reads, which is useful for filling gaps generated by complex repeat regions and low coverage regions in the genome assembly. Here, we used the TGS-GapCloser[82] (v 1.2.1) software and genomic HiFi data to fill the gaps generated in the assembly of the *P. australis* genome.

### RNA extraction and Full-length transcriptome sequencing
Total RNA was extracted from each of the nine tissue samples from *P. australis* using the Trizol method, including flower, stem apical meristem, aerial stems, leaves, aerial stem buds, rhizome internodal tissues, rhizome nodal meristem, rhizome buds, and fibrous roots. Firstly, *P. australis* tissue samples were ground in liquid nitrogen and then 1 ml of Trizol reagent was added, mixed thoroughly and left for 5 min. Next, add 200 uL of chloroform, shake vigorously for 30 s, and leave at room temperature for 10 min. centrifuge at 15,000 rpm for 15 s and then aspirate the supernatant into a new EP tube. Subsequently, add an equal volume of isopropanol (4 °C) and mix thoroughly. After standing for 10 min, centrifuge at 15,000 rpm for 10 min at 4 °C. Finally, the precipitate was washed twice with 75% ethanol (4 °C). After drying, add RNA-FREE water to dissolve the precipitate and store at −20 °C.

We mixed equal amounts of RNA from nine *P. australis* tissues as pooling samples for full-length transcriptome sequencing. The full-length cDNA was prepared using a SMARTer™ PCR cDNA Synthesis Kit (Takara Biotechnology, Dalian, China). The SMRTbell libraries were constructed with the Pacific Biosciences DNA Template Prep Kit 2.0. The Library quantification and size was measured using Qubit 3.0 Fluorometer (Life Technologies, Carlsbad, CA, USA) and Bioanalyzer 2100 system (Agilent Technologies, CA, USA). Subsequently, SMRT sequencing was performed on a PacBio Sequel II platform by Frasergen Bioinformatics Co., Ltd. (Wuhan, China).

The raw sequencing data was preprocessed using SMRTlink software, and the Iso-Seq analysis process was employed to obtain full-length transcript sequences. The parameters are as follows:

ccs *.subreads.bam *.ccs.bam --min-rq 0.9 --maxLength 15,000 --minLength 50 --minPasses 3;

lima *.ccs.bam IsoSeqPrimers.fasta *.fl.bam --isoseq --peek-guess;

isoseq3 refine *.fl.primer_5p--primer_3p.bam IsoSeqPrimers.fasta *.flnc.bam --require-polya;

isoseq3 cluster *.flnc.bam *.flnc.clustered.bam --verbose --use-qvs.

IsoSeqPrimers.fasta:
>primer_5p
AAGCAGTGGTATCAACGCAGAGTACATGGGG
>primer_3p
AAGCAGTGGTATCAACGCAGAGTAC

### Identification of telomeres, centromere and 5-methylcytosine
We used tidk (0.2.3)[83] to search the genome for telomeric repeat sequences from length 5 to length 12, and the most abundant repeat sequence units were counted and visualized. The identification of centromere was based on the method described by Shi et al.[25], where we filtered the *P. australis* genome for repetitive sequences contained in the genome (filtered condition: period >= 50, copies >= 2.0 To obtain 5-methylcytosine information from Pacbio HiFi sequencing data, we first generated reads with 5mC tags from HiFi reads using jasmine (v 2.0.0, https://github.com/PacificBiosciences/jasmine) and used IGV to find centromere sequences[84]. Next, soft clip mapped reads with methylation tags using pbmm2 (v 1.13.1, https://github.com/PacificBiosciences/pbmm2). Finally, information on the 5mC locus in the *P. australis* genome was analyzed using pb-CpG-tools (v2.3.2, https://github.com/PacificBiosciences/pb-CpG-tools) and visualized using IGV.

The 5mC identification parameters are as follows:
ccs subreads.bam hifi_reads.bam --hifi-kinetics;
jasmine hifi_reads.bam 5mc.hifi_reads.bam;
pbmm2 align genome.fasta 5mc.hifi_reads.bam 5mc.hifi_reads.pbmm2.align.bam –sort;
aligned_bam_to_cpg_scores --bam 5mc.hifi_reads.pbmm2.align.bam --output-prefix 5mc.hifi.pbmm2 --model pileup_calling_model.v1.tflite.

### Allopolyploid subgenome phasing
First, we used blast[85] (-evalue ≤ 1e-10, -num_alignments = 10) to identify paralogous homologous genes within the *P. australis* genome, orthologous homologous genes between *P. australis* and *Oryza sativa*, and orthologous genes between *Phragmites australis* and *Panicum virgatum*. Next, we identified homologous chromosome pairs in the genomes of different species by JCVI (v0.5.7, https://github.com/tanghaibao/jcvi/)[86] and WGDI (v0.6.1)[87] and determined the blocks of colinearity between genomes. Based on the colinearity results, we extracted single-copy genes in the homologous chromosomes between *Phragmites australis-Oryza sativa* and *Phragmites australis-Panicum virgatum*, respectively, for sequence splicing and genetic distance estimation. Finally, with the *Panicum virgatum* outgroup, we used RAxML (v.8.2.X)[88] software to construct phylogenetic trees for the chromosomes of the three species. We classified the *P. australis* chromosome, which is closer to the homologous chromosome of *Oryza sativa*, to subgenome A and the other homologous chromosome to subgenome B.

### Evaluation of assembled genomes
We used multiple software and assembly metrics to assess our assembled genomes' completeness, accuracy, and consistency. First, we evaluated the assembly quality of our genome by calculating Length, N50, L50, GC content, and other metrics using QUAST (Quality Assessment Tool for Genome Assemblies)[89]. Next, we mapped the genome survey and Iso-seq sequencing data to the assembled genome using minimap2 software and assessed the genome integrity[90]. The uniformity of sequencing coverage and contamination in the sequencing data are based on the ratio of reads, coverage, and the distribution of the GC content with the sequencing depth. Subsequently, we evaluated the accuracy and consistency of the genomes by calculating the genome consensus quality value (QV) values using Merqury[91] software. The short-read sequencing data were mapped to the assembled genome using the BWA[92], and variant calling was performed via the GATK (GATK: https://github.com/broadinstitute/gatk). Subsequently, the resulting VCF files were subjected to a filtration process aimed at identifying and removing low-quality or false-positive variants, thereby obtaining high-confidence SNPs (single nucleotide polymorphism) and Indels (insertion-deletion variant) information. In conclusion, the accuracy and completeness of the genome were evaluated based on the number and distribution of single-nucleotide polymorphisms (SNPs) and insertion-deletion mutations (Indels). We used BUSCO (Benchmarking Universal Single-Copy Orthologs)[93,94] based on the Embryophyta_odb10 gene set and CEGMA (Core eukaryotic genes mapping approach)[95] based on 248 ultra-conserved core eukaryotic genes (CEGs) to assess genomic accuracy and completeness. To more accurately assess the coherence of our assembled genomes, we first used the combination of LTR_Finder[96] (-D 15000 -d 1000 -L 7000 -l 100 -p 20 -C -M 0.9) and LTR_harvest[97] (-similar 90 -vic 10 -seed 20 -seqids yes -minlenltr 100 -maxlenltr 7000 -mintsd 4 -maxtsd 6 -motif TGCA -motifmis 1) to identify long terminal repeats (LTR) sequences in our assembled whole genomes and two sets of subgenomes, respectively, and then used LTR_retriever[98] (-u parameter using the evolutionary rate of rice: 1.3e-8) to integrate the results and to calculate the LTR Assembly Index (LAI)[99] and LTR density, and finally the genomic LAI was visualized by ggplot2[100]. The visualisation of the synteny and structural variations between two T2T *P. australis* genomes (PaCui.No1 and CN) was conducted using the GenomeSyn (v 1.2.6)[101].

## Gene prediction and functional annotation

To identify the repeat contents in the *P. australis* genome more accurately and comprehensively, we combined two strategies of homology-based prediction and ab initio prediction to identify repetitive sequences in the *P. australis* genome.

Homolog-based approach: Based on RepBase[102,103], a database of known repetitive sequences (http://www.girinst.org/repbase), we used RepeatMasker (v4.0.7, www.repeatmasker.org) and RepeatProteinMask software to predict sequences that are similar to known repetitive sequences. Sequences that are similar to known repeats. Ab initio approach: first, an ab initio repetitive sequence library (http://www.repeatmasker.org/RepeatModeler/) was constructed using RepeatModeler software and LTR-FINDER[96], and subsequently, an ab initio repetitive sequence library was constructed by the RepeatMasker software for repeat sequence prediction. In addition, we predicted the Tandem Repeat in the *P. australis* genome using TRF software[104] and identified the SSR sites present in the *P. australis* genome by MISA[105].

We accurately predicted the gene structure of the *P. australis* genome by combining three strategies. Firstly, we compared the coding protein sequence of *P. australis* related species (*Panicum virgatum*, *Setaria italica*, *Setaria viridis*, *Sorghum bicolor*) with the genome sequence, and obtained the gene structure information of homology-based prediction using Exonerate[106]. Next, we obtained gene structure information for transcriptome-based prediction by mapping the Iso-Seq data to the genome and using PASA[107] to determine the gene's shear sites and exon regions. Finally, we performed de novo prediction of gene structures by AUGUSTUS[108] and GlimmHMM[109] software. Integrating the above prediction results, we used MAKER2[110] to combine the gene structure information predicted by various methods into a non-redundant and more complete gene set. We used PASA[107] to update the gene structures with the transcriptome data.

Functional annotation information of the genes was obtained by homologous search of the predicted gene set in several databases, including SwissProt[111], NR, PFAM[112,113], GO[114], KEGG[115], InterPro[116], TrEMBL[117].

Non-coding RNA prediction: based on the structural characteristics of tRNAs, tRNAscan-SE[118] was used to find tRNA sequences in the genome; based on the highly conserved nature of rRNAs, rRNA sequences of closely related species can therefore be selected as reference sequences and rRNAs in the genome can be found by BLASTN comparison; in addition, using Rfam[119] family of covariance models to predict miRNA and snRNA sequence information using INFERNAL[120].

## Gene family and phylogenetic analysis

We collected and organized protein sequences from *P. australis* and 14 species, including *Arabidopsis thaliana*, *Aegilops tauschii*, *Brachypodium distachyon*, *Cleistogenes songorica*, *Dendrocalamus latiflorus* Munro, *Oryza sativa*, *Panicum hallii*, *Panicum virgatum*, *Pennisetum purpureum* Schum, *Setaria italica*, *Setaria viridis*, *Sorghum bicolor*, *Triticum aestivum*, and *Zea mays*, for gene family identification and evolutionary analysis.

First, we performed a comprehensive protein sequence blastp analysis (E-value ≤ 1e-5) using diamond[121]. Subsequently, the blastp results were clustered using OrthoFinder2[122] for immediate homologous gene finding and Orthogroup construction, and single-copy and multicopy gene families were obtained. To perform an accurate multi-species phylogenetic relationship analysis, we filtered the single-copy immediate homologous gene families shared by all species. We retained only those genes with amino acid lengths ≥100. Next, multiple sequence comparisons were performed separately for genes within each single-copy homologous gene family using MUSCLE[123]. Next, multiple sequence alignment results were integrated and converted into super-gene alignment in phylip format. Finally, a phylogenetic tree was constructed by Maximum Likelihood using RAxML[88] with *Arabidopsis thaliana* as the outgroup.

Based on obtaining time-corrected points based on fossil evidence at the TimeTree[124] website and in the literature, we used r8s[125] and the mcmctree program in the PAML[126] package (based on the Bayesian relaxed molecular clock approach) to estimate divergence times for species. According to the results of gene families and phylogenetic trees, CAFÉ[127] was used to predict the expansion or contraction of gene families in different species in each evolutionary branch. Based on obtaining organisms each sharing a single-copy orthologous homologous gene family member, we used the Codeml program in the PAML[126] package (using the branch-site model) to test whether the gene was under positive selection.

## Whole genome doubling (WGD) and karyotype evolution analysis

We performed blastp[85] comparisons between multiple species genomes to obtain orthologous pairwise (best comparison results for mutual blastp). Subsequently, The MCscan (https://github.com/tanghaibao/jcvi/)[86] was utilized to search for covariate segments between species genomes, and 4dTv values and Ks values were calculated for gene pairs contained in the covariate segments. We used the time of divergence of the evolutionary trees of *P. australis* and *Cleistogenes songorica* (Ks = 0.305, T = 34.6 mya, μ = 4.41e-08) to infer the evolutionary rate μ according to the formula Ks = 2 μT, after which we calculated the time of divergence between species and the time of occurrence of the species WGDs based on μ.

We updated the ancestral genome sequence of the Ancestral monocot karyotype except for Acoraceae (AMK-A) obtained from WGDI using the genetic information of *Sorghum bicolor*, *Triticum aestivum*, *Oryza sativa*, *Cleistogenes songorica*, *Phragmites australis*, *Panicum hallii*, and *Setaria viridis* by the parameter "-akr", and the genetic information in *P. australis* by the parameter "-km" visualization of genome karyotypes in *P. australis*.

## Transcriptome sequencing and WGCNA analysis

Sequencing libraries were constructed from RNA extracted from different tissues of *P. australis* (mature leaves, aerial stems, and rhizome tissues) and sequenced using the BGI T7 platform for PE150 short reads. After filtering by fastp (v0.21.0)[128], we mapped different tissue clean reads to the *P. australis* genome using HISAT2 (v2.2.1)[129] and quantified reads by featureCounts (v2.0.3)[130]. We used FPKM to indicate the expression value of each gene. We used FPKM to indicate the expression value of each gene. Subsequently, we performed a weighted gene co-expression network analysis of the average FPKM > 1 gene in all tissues using the WGCNA[131] (v1.72-1). Tissue-specific gene clusters were identified based on module-tissue correlations (Pearson | cor | > 0.95). Genes in the obtained tissue-specific modules were analyzed for GO and KEGG enrichment using the clusterProfiler (v4.0) package[132].

## Identification of Lugen polysaccharide biosynthetic pathway genes and prediction of transcription factors regulating *PaSUC* genes

We screened the structural genes related to polysaccharide biosynthetic pathways by filtering them in the functional annotation results of structural genes. Then, we reconfirmed the screening results by identifying specific structural domains in these structural genes by HMMER[133]. To obtain the Hub genes in the tissue-specific module, the genes with weight > 0.4 were further analyzed by the Radiality algorithm in the CytoHubba plugin. We defined the Top 100 genes as Hub genes and then visualized the Hub gene co-expression network using Cytoscape. Transcription factor prediction was performed by submitting the protein sequences of the genes in the module to the Plant TFDB (v5.0) database[134]. Cis-acting element prediction was performed by uploading the sequences within 2000 bp upstream of the *PaSUC* gene into the PlantCARE database[135]. Genomic circos and gene expression and cis-acting elements were visualized by TBtools (v2.03)[136].

## Extraction of polysaccharides from *phragmitis rhizome*

Fresh *P. australis* leaves, aerial stems, and rhizomes (with scale leaves and fibrous roots removed) were washed and chopped, then dried and crushed at 70 °C. Each 10 g of tissue powder was taken, added to 100 ml of ddH$_2$O and ultrasonicated at 80 °C, 250 W for 2.5 h. A total of three extractions were performed. The extract was then concentrated to 100 ml under reduced pressure, 4 times the volume of anhydrous ethanol was added, stirred and mixed, then precipitated for 24 h at 4 °C. The precipitate was collected and

washed three times in succession with anhydrous ethanol, ether and acetone. The washed precipitate was re-solubilised by addition of appropriate ddH$_2$O, dialysed (3500 MWCO) for 48 h and lyophilised ($-56\,°C$) to obtain crude polysaccharide. The concentration of polysaccharides was determined by measuring the absorbance of the samples at 490 nm using the Phenol-Sulfuric acid method. Protein impurities in the crude polysaccharide were determined by measuring the absorbance of the samples at 595 nm using the Coomassie Brilliant Blue G-250 method. Polysaccharide mass (mg) = crude polysaccharide mass (mg) * polysaccharide concentration (%).

### Dual-luciferase assays

The CDS sequence of PaR2R3-MYB was initially cloned using Phanta Max Super-Fidelity DNA Polymerase (Nanjing Vazyme Biotechnology Co., China) and subsequently inserted into the pGreenII 62-SK vector as an effector (pGreenII-62-SK-PaR2R3-MYB), in accordance with the instructions provided by the ClonExpress II One Step Cloning Kit (Nanjing Vazyme Biotechnology Co., China). Primer sequences used for vector construction are shown in Supplementary Fig. 23b. A candidate promoter sequence for *PaSUC1.1* with enzymatic cleavage sites (KpnI and BamHI) was obtained through synthetic synthesis (Suzhou Azenta Life Sciences Biotechnology Co., China) and ligated into the pGreenII 0800-LUC vector as a reporter (pGreenII-0800-LUC-*p*PaSUC1.1) using T4 DNA ligase (Suzhou Azenta Life Sciences Biotechnology Co., China). Subsequently, the constructed vectors were transformed into Agrobacterium GV3101(pSoup) (Shanghai Weidi Biotechnology Co., China), respectively. Next, Agrobacterium of pGreenII-62-SK-PaR2R3-MYB + pGreenII-0800-LUC-*p*PaSUC1.1 and pGreenII-62-SK + pGreenII-0800-LUC-*p*PaSUC1.1, respectively, were injected into the *N. benthamiana* on both sides of the same leaf for transient expression. Following the uniform spraying of infected *N. benthamiana* leaves with the substrate D-Luciferin potassium salt (Gold Biotechnology, Inc., USA), the presence of LUC fluorescence signals was detected utilising a chemiluminescence imaging system, namely the Tanon 5200 (Shanghai Tianneng Life Science Co., China). LUC and REN luciferase activities were assayed in a Thermo Scientific Varioskan Flash (Thermo, USA) according to the instructions of the Luciferase Assay System kit (Promega, USA).

In addition, the binding of PaR2R3-MYB to the *PaSUC1.1* promoter fragment (GAAACACAACTGATCCGAC / GTCGGATCAGTTGTGTT TC) was predicted using AlphaFold3 [137] with Ca$^{2+}$ and Mg$^{2+}$ added to the model.

### Statistics and reproducibility

The data in our study are available from the corresponding author to ensure reproducibility of the analyses. Three biological replicates were performed for each tissue in transcriptome sequencing ($n = 3$). In the polysaccharide extraction experiments, three biological replicates were set up for each tissue for polysaccharide extraction and determination of crude polysaccharide mass ($n = 3$). The crude polysaccharides obtained from each tissue were combined for polysaccharide content and protein content determination, and the biological replicates were set to 4 ($n = 4$). Gene function enrichment analysis was performed using ClusterProfile in this study. The statistical significance of the Gene Ontology terms was evaluated using Fisher's exact test in combination with the FDR correction for multiple testing, with a significance level of $P < 0.05$. All values are expressed as standard deviation ± mean. The statistical significance of the findings is based on a two-tailed *t*-test, with a P-value of less than 0.05 indicating statistical significance.

### Reporting summary

Further information on research design is available in the Nature Portfolio Reporting Summary linked to this article.

### Data availability

Whole genome and transcriptome raw sequencing data used in this study have been deposited at the National Center for Biotechnology Information (NCBI) under accession number PRJNA1055898. The genome assembly, annotations, protein sequences, CDS sequences, uncropped and unedited gel images, and the numerical source data underlying the graphs and charts are available for download from Figshare (https://doi.org/10.6084/m9. figshare.27016279). Meanwhile, the uncropped and unedited gel images were provided as Supplementary Fig. 24.

### Code availability

All software and pipelines were executed according to the official instructions. No custom code was generated for this study.

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

## Acknowledgements
This study was accomplished under the financial support of the National Natural Science Foundation of China (31972934, 31170784) and the Capital Normal University's Capacity Building of Science and Technology Innovation Service-Basic Scientific Research Operating Expenses (No.19530050183). We thank Xikun Wu and Zhiqiang Wang for their help in material collection. We would like to express our gratitude to Professor Liangyu Liu for kindly providing plasmid empty vectors for our experiments. Thanks to Wuhan Fraser Genetic Information Co. for help in genome sequencing and assembly.

## Author contributions
J.P.C. participated in the experimental design and was responsible for bioinformatics data analysis, experimental validation, Experimental validation, manuscript writing, writing review and editing. Manuscript proofreading and reference organization by R.W., R.Q.G., and M.H.C.; R.W., R.Q.G., and Z.Y.W. were responsible for the management and sampling of the plant samples. L.L. and J.M.H. participated in genomic survey analysis. SXC was responsible for this study's supervision, design, writing review and editing. All authors read and approved the final manuscript.

## Competing interests
The authors declare no competing interests.
