## [Transparent Peer Review file · Communications Biology]

Telomere-to-telomere *Phragmites australis* reference genome assembly with a B chromosome provides insights into its evolution and polysaccharide biosynthesis

Corresponding Author: Professor Suxia Cui

Version 0:

Reviewer comments:

Reviewer #1

(Remarks to the Author)

This article is presenting the phased chromosome scale genome assembly and annotation of *Phragmites australis*, a widely distributed Poaceae specie. Genome evolution and gene expression analyses are presented. The methods used are well described and the results seem of quality. The conclusions drawn by the authors are relevant. The results are presented in detail, which sometimes makes them a little difficult to read. Indeed, the results section is very dense and is less well structured than the other sections.

I have only few minor remarks:

- The sentence (lines 303-306) has to be rephrased
- I think the term "phased" is more appropriate than "disassembled" (line 479)
- Can you detail the "modified CTAB method" for HMW DNA extraction? These protocols are still sought after by laboratories wishing to extract HMW DNA from plants.
- The term "scaffolding" is more appropriate than "mounting" (line 640)
- Supp. Table 1 : "sketch" can be removed
- In the table S4, Iso_Seq data are presented but I didn't see any information on Iso_Seq data generation in the methods.
- RNA extraction protocol is not presented in the methods.

Reviewer #2

(Remarks to the Author)

The authors have performed an important service in presenting a high-quality reference genome, and B chromosome that is novelty. I hope authors further analyze the important of B chromosome, and validate genes or sequences using experiments. In the polysaccharides biosynthetic pathway, a PaMYB gene regulating PaSUC1.1 through coexpression analysis. However, I could not find more further results expect for data driven. More importantly, the authors should rephrase more results and polish the manuscript very carefully, and rewrite it.

1. As a reader or reviewer, I can't find more valuable information about *Phragmites australis* in the Introduction. The second should introduce the roles of B chromosomes, rather than the definition. The third and fourth paragraphs should be deleted.
2. The English needs some polishing in many places. For example, 'heterotetraploid' should be 'allotetraploid'. Chromosomes should be sorted in Figure 1B. Figure 2e, Figure 5b should be deleted.
3. The quality of English should be considerably improved prior to resubmission. The manuscript should be concise, not data driven.
4. Line 122-124. "Next, we searched telomeric ... a repeat number 40,863" should be removed. The AAACCCT motif is conserved in plants.
5. Line 164. Why *P. australis* has more gene with >5kb in length? How about the length of CDS and intron?
6. Line 203. Chromosome 25 should be B chromosome. Authors should check it carefully and use it with guidance. And I can't find two B chromosomes based on homologous analysis.
7. Line 247. Do authors have more evidence could support the second insertion events which maybe false?
8. Authors should merge 'Analysis of the gene family' and 'P. australis evolutionary position and whole genome duplication analyses'.
9. Coexpression analysis is an important strategy for the identification of gene. Except for three uniquely modules, other modules have also higher pearson cor values, including positive and negative. Please provide some explanations.

10. Authors need to provide experiments supporting the relationship between PaMYB and PaSUC1.1.

11. Too many results are found in the Discussion.

12. What is the evolutionary rate μ ?

Reviewer #3

(Remarks to the Author)

The T2T genome assembly of the *P. australis* genome is of great significance for our understanding of Arundiaceae evolution and polysaccharide biosynthesis in phragmites rhizoma, but the current manuscript content is somewhat simple, and the following points need to be stressed:

- 1、 Figure1 is not clear and needs to be modified.
- 2、 The value and application of assembling the B chromosome genome should be elaborated in the article.
- 3、 What method is used to filter out TFs in Fig6, and why is it necessary to analyze their expression differences
- 4、 The polysaccharide biosynthesis pathway is mentioned in Figure 6, and the differences in polysaccharide content and their accumulation patterns in different tissues should be supplemented.

Reviewer #4

(Remarks to the Author)

Cui et al. assembled the T2T genome of *Phragmites australis* and successfully assembled the B genome. I have rarely seen anyone able to assemble heterozygous aneuploid tetraploid, so I was interested to read this manuscript. However, I still have some issues that need to be clarified.

1. The biggest problem is that there is Ultra-long ONT sequencing data, and I believed that HiFi data is not enough to assemble the genome of such a complex polyploidy plant with T2T level. So, I hope that the authors would like to explain and describe this in detail, which is a challenge even for now.
2. The authors successfully assembled a genome with very few gaps, so why not try to fill them with a little more efforts by other skills
3. For the successful assembly of the B genome, it seems that the result is not particularly satisfactory, what is the author's explanation or means of improvement.
4. Line 148, How are SNPs and indels calculated on a reference genome? What is the purpose of doing this.

There are some minor suggestions

Line 114 "482.94 Raw data" inappropriate words.

Line 142 "survey" Inappropriate words

Line 177 "sequenced plant" instead of Latin name, and Lack of scale in the figure

Line 178 Latin name italicized.

Version 1:

Reviewer comments:

Reviewer #2

(Remarks to the Author)

The authors have addressed most of my concerns, and the language has been improved a lot. There are still some minor issues to be addressed:

- 1.Line 116-117, 229: "The obtained chromosome-level genomes had a gap on Chr21 and Chr25, respectively. Subsequently, the gap on the Chr21 chromosome was successfully filled using TGS-GapCloser software and HiFi data". But in figure1B, 50 chromosomes named Chr1A-12A, Chr1B-12B, and ChrB, which are not consistent with line 229 as well. Please confirm.
- 2.Figure1 needs to be modified, the Chromosomes are not legible.
- 3.The Chromosome names should be consistent. For example, in line 162, the chromosome labels have no space between the 'Chr' and the number (Chr7B, Chr7A, and Chr5B), but in line 229, there is a space between the 'Chr' and the number (Chr 9, Chr 12, Chr 19, and Chr 20).
- 4.The language needs to be more concise.

Reviewer #3

(Remarks to the Author)

It has been revised accordingly, and can be accepted.

Version 2:

Reviewer comments:

Reviewer #2

(Remarks to the Author)

The comments and suggestion of reviewers have been well responses and employed in the revised manuscript, thus, I suggest to accept it for publication.

Dear Editors and Reviewers:

Thanks for your and the reviewer's comments concerning our manuscript entitled “**Telomere-to-telomere *Phragmites australis* reference genome assembly with a B chromosome provides new insights into its evolution and polysaccharide biosynthesis**”. Those comments are all valuable and helpful for revising and improving our manuscript, as well as the essential guiding significance to our research. We have studied comments carefully and have made corrections point-by-point, which we hope meet with approval. Furthermore, given that the growth period of the *P. australis* material employed in the supplementary experiments predominantly spans the months of June to September, I extend my sincerest apologies for being unable to return the revised manuscript promptly.

Reviewer #1 (Remarks to the Author):

This article is presenting the phased chromosome scale genome assembly and annotation of Phragmites australis, a widely distributed Poaceae specie. Genome evolution and gene expression analyses are presented. The methods used are well described and the results seem of quality. The conclusions drawn by the authors are relevant. The results are presented in detail, which sometimes makes them a little difficult to read. Indeed, the results section is very dense and is less well structured than the other sections.

I have only few minor remarks:

- The sentence (lines 303-306) has to be rephrased

Response: We are grateful for the positive comments and have amended the sentence accordingly (Lines 360-364).

Lines 360-364: The GO functional enrichment results indicated that these expanded gene families were primarily associated with the functions of telomere maintenance (GO:0032200, GO:0010833, GO:0007004, and GO:0000723), DNA binding (GO:1990837, GO:0043565, GO:0000976, and GO:0003690), and starch synthesis (GO:0004556 and GO:0016160) (Figure S16c).

- I think the term “phased” is more appropriate than “disassembled” (line 479)

Response: Thank you for your correction. We have carefully revised the manuscript, and the changes can be found in line 594.

- Can you detail the “modified CTAB method” for HMW DNA extraction? These protocols are still sought after by laboratories wishing to extract HMW DNA from plants.

Response: According to the comment. We have added detailed steps for DNA extraction in the Materials and methods section (Lines 762-779).

Lines 762-779:

DNA extraction

High-quality genomic DNA was extracted from young leaves of *Phragmites australis* using a modified CTAB method. First, we added 5 ml of 2% CTAB, 50 μ L of β -mercaptoethanol, and 26 μ L of proteinase K to *P. australis* leaf tissues after sufficient grinding (with the addition of PVP), mixed thoroughly, and incubated at 65°C for 30 min. After centrifugation at 12000 rpm for 10 min, the supernatant was retained. Next, an equal volume of phenol/chloroform/isoamyl

alcohol (25:24:1) was added and mixed thoroughly. After centrifugation at 12000 rpm for 10 min (repeated 2 times), and retain the supernatant. Continue by adding an equal volume of chloroform/isoamyl alcohol (24:1) extracted once and centrifuged at 120,000 rpm for 10 min and retained the supernatant. Subsequently, pre-cooled isopropanol (4°C) was added, mixed upside down and allowed to stand for 30 min at 4°C. The precipitate was centrifuged at 12000 rpm for 10 min and retained. After washing twice by adding 1 ml of pre-cooled 70% ethanol at 4°C, the precipitate was dried at 37°C. Finally, the precipitate was dissolved with Tris-HCl and RNaseA and incubated at 37°C for 30 min, then stored at -20°C. The quality and quantity of the extracted DNA were examined using a NanoDrop 2000 spectrophotometer (NanoDrop Technologies, Wilmington, DE, USA), Qubit dsDNA HS Assay Kit on a Qubit 3.0 Fluorometer (Life Technologies, Carlsbad, CA, USA) and electrophoresis on a 0.8% agarose gel, respectively.

- The term “scaffolding” is more appropriate than “mounting” (line 640)

Response: Thank you for your correction. We have carefully revised the manuscript; the changes can be found in line 807.

- Supp. Table 1: “sketch” can be removed

Response: We have carefully reviewed this and thank you for your correction.

- In the table S4, Iso_Seq data are presented but I didn't see any information on Iso_Seq data generation in the methods.

- RNA extraction protocol is not presented in the methods.

Response: We are grateful for your valuable correction. We have incorporated detailed instructions for the RNA extraction and processing of full-length transcriptome data in the Materials and Methods section (Lines 812-843).

Lines 812-843:

RNA extraction and Full-length transcriptome sequencing

Total RNA was extracted from each of the nine tissue samples from *P. australis* using the Trizol method, including flower, stem apical meristem, aerial stems, leaves, aerial stem buds, rhizome internodal tissues, rhizome nodal meristem, rhizome buds, and fibrous roots. Firstly, *P. australis* tissue samples were ground in liquid nitrogen and then 1 ml of Trizol reagent was added, mixed thoroughly and left for 5 min. Next, add 200 μ L of chloroform, shake vigorously for 30 s, and leave at room temperature for 10 min. centrifuge at 15,000 rpm for 15 s and then aspirate the supernatant into a new EP tube. Subsequently, add an equal volume of isopropanol (4°C) and mix thoroughly. After standing for 10 min, centrifuge at 15000 rpm for 10 min at 4°C. Finally, the precipitate was washed twice with 75% ethanol (4°C). After drying, add RNA-FREE water to dissolve the precipitate and store at -20°C.

We mixed equal amounts of RNA from nine *P. australis* tissues as pooling samples for full-length transcriptome sequencing. The full-length cDNA was prepared using a SMARTer™ PCR cDNA Synthesis Kit (Takara Biotechnology, Dalian, China). The SMRTbell libraries were constructed with the Pacific Biosciences DNA Template Prep Kit 2.0. The Library quantification and size was measured using Qubit 3.0 Fluorometer (Life Technologies, Carlsbad, CA, USA) and Bioanalyzer 2100 system (Agilent Technologies, CA, USA). Subsequently,

SMRT sequencing was performed on a PacBio Sequel II platform by Frasergen Bioinformatics Co., Ltd. (Wuhan, China).

The raw sequencing data was preprocessed using SMRTlink software, and the Iso-Seq analysis process was employed to obtain full-length transcript sequences. The parameters are as follows:

```
ccs *.subreads.bam *.ccs.bam --min-rq 0.9 --maxLength 15,000 --minLength 50 --minPasses 3;
```

```
lima *.ccs.bam IsoSeqPrimers.fasta *.fl.bam --isoseq --peek-guess;
```

```
isoseq3 refine *.fl.primer_5p--primer_3p.bam IsoSeqPrimers.fasta *.flnc.bam --require-polya;
```

```
isoseq3 cluster *.flnc.bam *.flnc.clustered.bam --verbose --use-qvs.
```

IsoSeqPrimers.fasta:

```
>primer_5p
```

```
AAGCAGTGGTATCAACGCAGAGTACATGGGG
```

```
>primer_3p
```

```
AAGCAGTGGTATCAACGCAGAGTAC
```

Reviewer #2 (Remarks to the Author):

The authors have performed an important service in presenting a high-quality reference genome, and B chromosome that is novelty. I hope authors further analyze the important of B chromosome, and validate genes or sequences using experiments. In the polysaccharides biosynthetic pathway, a PaMYB gene regulating PaSUC1.1 through coexpression analysis. However, I could not find more further results expect for data driven. More importantly, the authors should rephrase more results and polish the manuscript very carefully, and rewrite it.

1. *As a reader or reviewer, I can't find more valuable information about Phragmites australis in the Introduction. The second should introduce the roles of B chromosomes, rather than the definition. The third and fourth paragraphs should be deleted.*

Response: Thank you so much for your valuable suggestions! Based on your comments, we've made some changes to the Introduction section (Lines 43-49, Lines 64-77).

Lines 43-49: *Phragmites australis* are widely distributed in rivers, lakes, dunes, alkaline salt flats, and other habitats worldwide, with strong environmental adaptability and colossal biomass. The high cellulose content of *P. australis* stalks can be used as an excellent raw material for paper production and as pasture for large animals. Moreover, *P. australis*' complex and extensive root network can absorb many kinds of heavy metal ions, and is widely used for water purification, wetland protection and soil stabilisation.¹ It has essential ecological and economic values in ecological protection, animal fodder, and traditional Chinese medicine.

Lines 65-78: With the continuous improvement and development of High fidelity reads sequencing (HiFi) and high-throughput chromosome conformation capture (Hi-C) technologies, the resolution of B chromosome (Bs) has been significantly improved¹⁰. Bs are often thought not to have any function, but large-scale histological studies are now characterizing the specific functions of Bs in certain species^{11,12}. As B chromosomes have been discovered and assembled in more species, their functions have been more deeply analyzed¹³⁻²⁰. The presence of Bs can affect the phenotype and reproduction of organisms, particularly affecting pollen formation and

fertilization in plants. The Bs can preferentially attach to the spindle on the egg side in female meiosis to segregate and produce unequal gametes¹³⁻¹⁵. Bs was observed to have nondisjunction during the first pollen mitosis in *Secale cereal* and *Aegilops speltoides*. Moreover, Bs in maize were observed to undergo nondisjunction in the second meiotic division, and sperm nuclei with two Bs would preferentially fertilize the egg cell²¹. In addition, active genes or non-coding RNAs in Bs can also affect maize A-chromosome (As) gene transcriptional profiles or phenotypes¹⁸⁻²⁰.

2. *The English needs some polishing in many places. For example, 'heterotetraploid' should be 'allotetraploid'. Chromosomes should be sorted in Figure 1B. Figure 2e, Figure 5b should be deleted.*

Response: Thank you very much for your corrections. We have scrutinized the manuscript and revised the images and vocabulary based on your comments (Figure 1B, Figure 2, and Figure 5).

3. *The quality of English should be considerably improved prior to resubmission. The manuscript should be concise, not data driven.*

Response: Thanks to your valuable comments. We have carefully adjusted and revised the manuscript to make it more concise.

4. *Line 122-124. "Next, we searched telomeric ... a repeat number 40,863" should be removed. The AAACCCT motif is conserved in plants.*

Response: Thank you for your correction. Based on your comments, we have revised the manuscript (Lines 117-119).

Lines 117-119: All chromosomal telomeres (AAACCCT) were successfully assembled, including 19 chromosomes with telomeres assembled at both ends (Tables S2 and S3).

5. *Line 164. Why P. australis has more gene with >5kb in length? How about the length of CDS and intron?*

Response: Thank you for your careful review, which is very important for our research. We have added further analyses and descriptions of CDS and intron lengths in the *P. australis* genome (Lines 178-203).

Lines 178-203:

Introns in the *P. australis* genome have driven increased gene lengths

Compared to Poaceae relatives, there was a notable decline in the proportion of genes with lengths below 2000 bp, while the proportion of genes exceeding 5000 bp increased significantly in *P. australis* (Figure 2a). In contrast, the overall distribution of introns, exons, and CDS lengths of *P. australis* was not significantly different from that of the related species. However, the proportion of genes containing introns was significantly higher at 89.91% (Figure 2b, Table S6). Subsequently, a more comprehensive structural characterization of the 11,654 genes in *P. australis* with a length exceeding 5,000 bp was conducted (Figure 2c). The results demonstrated that the genes with more than ten introns constituted 29.46% (3433) of these genes. The lengths of introns and exons exhibited a concentration in the range of 3000 bp to 5000 bp and 1 bp to 3000 bp, respectively, which were markedly higher than their respective mean values (mean

length of introns: 720.66 bp, mean length of exons: 301.18) (Table S6). A strong positive correlation was observed between intron length and gene length for these genes (Pearson $R = 0.99$, $p < 2.2e-16$) (Figure 2d and 2e). It is noteworthy that 2,473 genes with ultra-long introns (introns exceeding 10 kilobases in length) were identified in *P. australis*. The ultra-long introns did not appear to exert a deleterious effect on gene expression. In addition, certain genes exhibited significantly elevated tissue-specific expression (Figure 2f). The utilization of full-length transcriptome data from the PacBio platform for assisted annotation and optimization of gene structures has resulted in a notable enhancement in the accuracy and comprehensiveness of the gene annotations. Consequently, it is unlikely that these gene structures with ultra-long introns result from annotation errors. The accuracy of the structural annotation of these genes was confirmed by mapping the RNA-seq and Iso-seq data to representative genes with tissue-specific expression (with ultra-long introns) (Figure S8). Furthermore, the introns of these genes contained a substantial number of transposon sequences (Figure S8), which may have contributed to the formation of these ultra-long introns, thereby driving the increase in gene length.

6. *Line 203. Chromosome 25 should be B chromosome. Authors should check it carefully and use it with guidance. And I can't find two B chromosomes based on homologous analysis.*

Response: Indeed, we observed this chromosome's homology phenomenon, which has also been documented in studies of the maize B chromosome (Lines 233-239). Due to the limitations of current sequencing technologies and the insufficient sequencing depth achieved in this study, two sets of haplotype genomes of *P. australis* were assembled, comprising only one B chromosome. To confirm the accuracy of the chromosome assembly, we employed the FISH and Hi-C technology (Fig. 1a and Fig. S4a) to verify the presence of Chr 25. Furthermore, we utilized the BGI platform for short-read sequencing data and PCR technology to confirm the continuity of the assembled chromosome (Fig. S4e and Fig. S11). Furthermore, the B chromosome is characterized in greater detail in the Results and Discussion sections, respectively (Lines 623-647).

Lines 233-239:

This distinctive co-linearity outcome was similarly documented in the maize genome B chromosome investigation, which may represent a distinctive attribute of the B chromosome²⁹. The quality and integrity of the Chr25 chromosome assembly were verified by sequencing depth and coverage of short-read sequencing data (Figure S4e and S11a). Furthermore, the continuity of the assembly of a high-density region (10 MB in length) of tRNA on this chromosome was confirmed by PCR experiments (Figure S11b).

Lines 623-647:

Here, we provide a detailed characterisation of the assembled B chromosome: (1) Similar to Bs in maize¹⁰, most of the genes in Bs of *P. australis* are highly homologous to those in the A chromosome. However, larger blocks of colinearity could not be detected (Figure 3). (2) 87.88% of the sequences in the Bs are transposable sequences. Furthermore, the evolutionary analysis of LTR-RTs indicates that these transposable elements are closely related to those in the A chromosome (Figure S12). (3) Multiple candidate centromere regions identified in Bs, which predicts that Bs has undergone multiple breakage, fusion and chromosomal structural

rearrangement events (Figure S3). (4) The high density of repetitive and transposable sequences within the chromosomal regions of Bs may have played a role in mediating chromosome breakage and fusion events (Figures 1b and S3). (5) Bs of *P. australis* serve to prevent genomic instability by maintaining the heterochromatin state and transposon silencing through the establishment of a considerable number of methylation sites (5-mC) (Figure S3). Based on the aforementioned results, we postulate a possible mode of Bs production in *P. australis*: the breakage of the centromere region of the A chromosome, which results in fragmented chromosomes that undergo fusion and recombination to form a multimeric B chromosome. This process is mediated by centromere repeats and transposable sequences, or telomeric repeats, and is thought to play a significant evolutionary role. Subsequently, following gene rearrangement and selection during long-term evolution, this Bs retains genes that are beneficial for functional maintenance and structural stability, and maintains the stable state of the chromosome centromere structure and the transmission of genetic information through extensive methylation. The assembly and study of the *P. australis* B chromosome will provide a valuable genetic resource for elucidating the intricacies and diverse array of ecological adaptations of the *P. australis* genome. Moreover, it will serve as a distinctive model for investigating chromosomal genetics, gene regulation, and the mechanisms of chromosome evolution.

7. *Line 247. Do authors have more evidence could support the second insertion events which maybe false?*

Response: Thank you for your attention to this important issue. We provided new evidence for two insertion events by evolutionary analyses of LTR-RT sequences (Lines 301-308) and modified Figure 4 and Figure S15.

Lines 301-308:

We conducted evolutionary analyses of Copia and Gypsy sequences extracted during the two insertion events, respectively. The results demonstrated that a notable expansion of the Gypsy family occurred during the ancient insertion event. This result is consistent with the evidence that the Gypsy family experienced two periods of significant expansion, as illustrated in Figures 4c and 4d. The large number of insertions of these elements may affect gene sequences in the vicinity of TEs to a certain extent, providing abundant raw material for variation in genome evolution.

8. *Authors should merge 'Analysis of the gene family' and 'P. australis evolutionary position and whole genome duplication analyses'.*

Response: Thanks for the positive comments. We have merged and revised these two parts (Line 339).

9. *Coexpression analysis is an important strategy for the identification of gene. Except for three uniquely modules, other modules have also higher pearson cor values, including positive and negative. Please provide some explanations.*

Response: We performed functional enrichment on $|\text{Pearson cor}| > 0.8$ modules, and added descriptions of gene function enrichment results in these modules (Lines 464-480).

Lines 464-480: There was also a correlation between certain modules and organisations

(Pearson $cor| > 0.80$). A significant negative correlation (Pearson $cor = -0.81$) with leaf blades was observed for genes in the MEpink module mainly involved with microtubules or the cytoskeleton (Figure S19a). The down-regulation of the expression of these genes not only reduced the mechanical stiffness of the cell wall but also the maintenance of cellular tension and expansion pressure, resulting in a softer leaf morphology. *P. australis* is typically a densely tufted tall graminoid, and the softer leaves facilitate the capture of scattered and low-angle light in the presence of external forces such as wind and gravity. The MEyellow module demonstrated a significant positive correlation with aerial stems (Pearson $cor = 0.84$), as well as a significant negative correlation with rhizomes (Pearson $cor = -0.86$) (Figure S19b). The genes in MEyellow are primarily associated with ribosome biogenesis, rRNA processing, protein synthesis, and protein folding-related functions. In comparison to the rhizome, which is a nutritive organ, the aerial stems required large amounts of proteins to carry out functions such as water and nutrient transport, response to environmental stimuli and signalling. The rapid growth of *P. australis* places high demands on the correct folding and assembly of newly produced proteins in the aerial stems. The high level of gene expression in the MEyellow module ensures fast, accurate and efficient protein synthesis to meet the needs of rapid growth.

10. *Authors need to provide experiments supporting the relationship between PaMYB and PaSUC1.1.*

Response: To demonstrate the binding of PaMYB to the *PaSUC1.1* promoter, we performed dual luciferase reporter gene (dual-LUC) experiments, and the results were added to lines 553-565.

Lines 553-565:

The binding probability of the MBS element to the PaR2R3-MYB was then predicted using AlphaFold3 (ipTM = 0.89, pTM = 0.40) (Figure S21). To determine the effect of PaR2R3-MYB on the transcriptional regulation of *PaSUC1.1*, we performed dual luciferase reporter gene (dual-LUC) experiments in *Nicotiana benthamiana* using the PaR2R3-MYB amino acid coding sequence and a sequence 2000 bp upstream of the *PaSUC1.1* gene (Figure 7e and Figure S22). The pGreenII-62-SK-PaR2R3-MYB (35S::PaMYB) expression vector was employed as the effector, while the pGreenII-0800-LUC-*pPaSUC1.1* (*pPaSUC1.1*::LUC) expression vector, ligated to the *PaSUC1.1* candidate promoter, served as the reporter. In vivo imaging results showed that *N. benthamiana* tissues injected with 35S::PaMYB + *pPaSUC1.1*::LUC displayed stronger chemiluminescent signals compared to empty vector. Further dual luciferase activity assays showed that the relative activity of firefly luciferase (LUC/REN) was significantly elevated in *N. benthamiana* tissues injected with the combination of 35S::PaMYB + *pPaSUC1.1*::LUC (Figure 7e).

11. *Too many results are found in the Discussion.*

Response: Thank you for your correction. The discussion section was redacted with descriptions of results that were not deemed to be significant.

12. *What is the evolutionary rate μ ?*

Response: The calibration of the time of species divergence using multiple time points of fossil evidence in the phylogenetic tree construction resulted in a more accurate separation time of

34.6 mya for *Phragmites australis* and *Cleistogenes songorica*. This enabled the inference of an evolutionary rate (μ) of 4.41e-08 based on $K_s = 2 \mu T$. (Line 963)

Reviewer #3 (Remarks to the Author):

The T2T genome assembly of the P. australis genome is of great significance for our understanding of Arundiacae evolution and polysaccharide biosynthesis in phragmites rhizoma, but the current manuscript content is somewhat simple, and the following points need to be stressed:

1. *Figure 1 is not clear and needs to be modified.*

Response: Thanks for the positive comments. We have redrawn Figure 1 to make it clearer.

2. *The value and application of assembling the B chromosome genome should be elaborated in the article.*

Response: The Results and Discussion section for the *P. australis* B chromosome has been revised and expanded with new descriptions pertaining to the utility and applications of the B chromosome (Lines 643-647).

Lines 643-647: The assembly and study of the *P. australis* B chromosome will provide a valuable genetic resource for elucidating the intricacies and diverse array of ecological adaptations of the *P. australis* genome. Moreover, it will serve as a distinctive model for investigating chromosomal genetics, gene regulation, and the mechanisms of chromosome evolution.

3. *What method is used to filter out TFs in Fig6, and why is it necessary to analyze their expression differences.*

Response: In response to your comment, we have included a description of the TF screening method in the results (Lines 542-546). WGCNA analysis is based on the clustering of expression similarities between genes, employing a correlation matrix to calculate a weighted neighbor-joining matrix to construct gene co-expression networks and identify gene modules. The differential expression analysis represents a crucial phase in the process of validating and screening transcription factors. The analysis of whether a candidate transcription factor exhibits significant expression changes under a specific tissue or phenotype helps to confirm its potential function (e.g., up-regulation or down-regulation of a target gene) in a particular biological context. Furthermore, the degree of differential expression of genes (e.g., fold change or p-value) can be employed as an additional criterion for the screening of transcription factors that are more likely to play a role under a particular tissue or phenotype. These transcription factors with a similar trend of differential expression between them and the target gene are more likely to be biologically significant.

Lines 542-546: In order to identify potential transcription factors regulating the expression of *SUC* genes in the MEbrown module, a screening of the top 100 hub genes was conducted using the Radiality plugin in CytoHubba. Genes with weights lower than 0.40 were filtered in order to ensure the accuracy of the results. Furthermore, the transcription factors in the hub gene co-expression network were identified based on the Plant TFDB (v5.0) database.

4. The polysaccharide biosynthesis pathway is mentioned in Figure 6, and the differences in polysaccharide content and their accumulation patterns in different tissues should be supplemented.

Response: In response to the comments provided, an extraction and determination of polysaccharides was conducted on *P. australis* leaves, aerial stems, and rhizomes using the ethanol subsiding method. The reasons for the observed differences in accumulation were also analyzed in connection with the GO enrichment results (Lines 453 - 463).

Lines 453 – 463: Here, we extracted and determined the polysaccharide content of different tissues of *P. australis* using the 'ethanol subsiding method'. The findings revealed that among the diverse tissues of *P. australis*, the rhizomes exhibited the highest polysaccharide content, reaching 25.65 mg/g (Table 2). Notwithstanding the observation of a greater quantity of crude polysaccharide precipitation in the leaves, the polysaccharide content was found to be exceedingly low (2.04 mg/g.WD), manifesting as a brown powder (Figure 6g). This phenomenon may be attributed to the fact that a multitude of intricate physiological processes (e.g., photosynthesis, respiration, metabolic activities, etc.) occurring in the leaves result in the production of a considerable quantity of substances, including polyphenols and pigments (Figure 6e). These metabolites were co-precipitated with polysaccharides when treated with 80% ethanol and oxidized to a brown color during extraction and lyophilization.

Reviewer #4 (Remarks to the Author):

Cui et al. assembled the T2T genome of Phragmites australis and successfully assembled the B genome. I have rarely seen anyone able to assemble heterozygous aneuploid tetraploid, so I was interested to read this manuscript. However, I still have some issues that need to be clarified.

1. *The biggest problem is that there is Ultra-long ONT sequencing data, and I believed that HiFi data is not enough to assemble the genome of such a complex polyploidy plant with T2T level. So, I hope that the authors would like to explain and describe this in detail, which is a challenge even for now.*

Response: Thank you for your detailed review. PacBio's HiFi (High-Fidelity) sequencing technology combines the advantages of long reads and high accuracy (>99.9%) to resolve better complex repetitive sequences and structural variants, which are critical for polyploid genome assembly. In recent years, there have been several cases of successful assembly of complex plant genomes using PacBio HiFi data, including some complex polyploid species. Here, we successfully assembled 25 chromosomes of *P. australis* based on HiFi and Hi-C sequencing data using the hifiasm's-i3 model, a model designed for complex polyploid genomes (e.g., triploid and above). Preliminary assembly results showed only one gap in chromosomes Chr21 and Chr25, respectively, with the rest of the chromosomes consisting of one Contig.

Reported using Pacbio HiFi sequencing with HiC-assisted assembly:

He S, Weng D, Zhang Y, et al. A telomere-to-telomere reference genome provides genetic insight into the pentacyclic triterpenoid biosynthesis in *Chaenomeles speciosa*. *Hortic Res.* 2023;10(10):uhad183.

Huang P, Li Z, Wang H, et al. A genome assembly of decaploid *Houttuynia cordata* provides insights into the evolution of *Houttuynia* and the biosynthesis of alkaloids. *Hortic Res.* 2024;11(9):uhae203.

Guo L, Xie F, Huang X, Luo Z. A chromosome-level genome of 'Xiaobaixing' (*Prunus armeniaca* L.) provides clues to its domestication and identification of key bHLH genes in amygdalin biosynthesis. *Plants (Basel)*. 2023;12(15):2756.

Qu C, Zhu M, Hu R, et al. Comparative genomic analyses reveal the genetic basis of the yellow-seed trait in *Brassica napus*. *Nat Commun.* 2023;14(1):5194.

Liu F, Zhao J, Sun H, et al. Genomes of cultivated and wild *Capsicum* species provide insights into pepper domestication and population differentiation. *Nature Communications*, 2023, 14(1): 5487.

Hong Z, Peng D, Tembrock LR, et al. Chromosome-level genome assemblies from two sandalwood species provide insights into the evolution of the Santalales. *Commun Biol.* 2023;6(1):587.

2. *The authors successfully assembled a genome with very few gaps, so why not try to fill them with a little more efforts by other skills*

Response: We would like to express our gratitude for your review. The gaps in chromosome 21 were successfully repaired using TGS-GapCloser software and genomic HiFi data (Line 116-117).

Line 116-117: Subsequently, the gap on the Chr21 chromosome was successfully filled using TGS-GapCloser software and HiFi data.

3. *For the successful assembly of the B genome, it seems that the result is not particularly satisfactory, what is the author's explanation or means of improvement.*

Response: The presence of this chromosome was verified with assembly continuity by short reads data (Figure S4e and Figure S11). However, the substantial number of repetitive sequences present in the *P. australis* B chromosome represents a significant challenge for complete assembly. In particular, the long segmental repetitive regions, including transposons and tandem repetitive sequences, contribute to the difficulty of gap-filling. Currently, B chromosome studies in maize employ flow-sorting techniques to distinguish B chromosomes from other chromosomes. Additionally, Oxford Nanopore data are used to generate more complete assemblies of B chromosomes. However, as the assembly of the B chromosome is still in its preliminary stages, there are limited reference sequences available for comparison. In the future, we intend to perform resequencing and complete assembly of the B chromosome to gain insights into its potential biological functions and genetic patterns.

Blavet N, Yang H, Su H, et al. Sequence of the supernumerary B chromosome of maize provides insight into its drive mechanism and evolution. *Proc Natl Acad Sci USA.* 2021;118(23):e2104254118.

4. *Line 148, How are SNPs and indels calculated on a reference genome? What is the purpose of doing this.*

Response: For a high-quality genome assembly, the mapped raw sequencing data should have

a high concordance (few mismatches and variants). This study examined single nucleotide polymorphism (SNP) and insertion/deletion (InDel) information in the *P. australis* genome based on short reads sequencing data. We then counted the frequencies and distributions of SNPs and Indels across the entire genome. Our findings revealed that the SNPs and Indels accounted for 0.97% and 0.06% of the *P. australis* genome, respectively (Table S5). Additionally, we observed that they were relatively evenly distributed across the chromosomes (Fig. 1b). The reasonable distribution of variants in the *P. australis* genome and the high ratio consistency indicate that the assembled genome is of ultra-high continuity and integrity (PaCui.No1).

There are some minor suggestions:

Line 114 “482.94 Raw data” inappropriate words.

Response: Thank you for your careful review. We have changed ‘Raw data’ to ‘subreads data’ (Line 108).

Line 108: We obtained 482.94 Gb **subreads data** and 32.53 Gb HiFi reads data using PacBio Sequel II platform genome sequencing.

Line 142 “survey” Inappropriate words

Response: Thank you for your careful review. We have changed ‘survey’ to ‘genome survey sequencing’ (Line 112).

Line 112: The genome sketch features were similar to the flow cytometry and **genome survey sequencing** analysis results.

Line 177 “sequenced plant” instead of Latin name, and Lack of scale in the figure

Response: Thank you for your careful review. We have changed ‘sequenced plant’ to ‘the *P. australis*’ and added Scale bars to the figure.

Line 178 Latin name italicized.

Response: I am grateful for your meticulous examination of the material. A meticulous examination of the nomenclature format for the species in question has been conducted.

Dear Editors and Reviewers:

Thanks for your and the reviewer's comments concerning our manuscript entitled "**Telomere-to-telomere *Phragmites australis* reference genome assembly with a B chromosome provides new insights into its evolution and polysaccharide biosynthesis**". Those comments are all valuable and helpful for revising and improving our manuscript, as well as the essential guiding significance to our research. We have studied comments carefully and have made corrections point-by-point, which we hope meet with approval. In accordance with your request and that of the reviewers, we have incorporated comparative analyses and descriptions with those previously reported for the *P. australis* genome (doi.org/10.1038/s42003-024-06660-1). Please refer to Table 1, lines 143-149, lines 629-636, and Figure S5 for details. Furthermore, we have incorporated some of the responses to Reviewer #4's comment 3 into the Discussion section, see Lines 660-669.

Lines 143-149: As demonstrated in Table 1, an evaluation of the documented assembly metrics for the *P. australis* genomes indicates a notable enhancement in the quality of the PaCui.NoI assembly^{13,32}. Furthermore, synteny analysis with the recently reported *P. australis* T2T genome (CN) demonstrated that, with the exception of a large structural variation (Inversion) between certain chromosomes, such as Chr B, Chr 1B, Chr 2B, Chr3B, and Chr 4B, the two genomes exhibited a high degree of conservation and consistency in the majority of structural regions (Figure S5).

Lines 629-636: A significant inversion in structural variation is observed between our assembled *P. australis* B chromosome and the recently reported B chromosome (Chinese lineage, CN) (Figure S5). However, the gene clusters that they have retained in evolution demonstrate high conservation in terms of functions relevant to the maintenance of chromosome stability and meiosis¹³. This coexistence of functional conservatism and structural variation will provide crucial insights for comprehensive investigations into the evolutionary implications of genomic structural variation, environmental adaptations, and genetic variations among *P. australis* strains.

Lines 660-669: However, the substantial number of repetitive sequences present in the *P. australis* B chromosome represents a significant challenge for complete assembly. In particular, the long segmental repetitive regions, including transposons and tandem repetitive sequences, contribute to the difficulty of gap-filling. Currently, B chromosome studies in maize employ flow-sorting techniques to distinguish B chromosomes from other chromosomes. Additionally, Oxford Nanopore data are used to generate more complete assemblies of B chromosomes. However, as the assembly of the B chromosome is still in its preliminary stages, there are limited reference sequences available for comparison. As more *P. australis* genomes or B chromosomes are resequenced and completely assembled in the future, they will provide distinctive models for the study of chromosome genetics and chromosome evolutionary mechanisms.

Reviewer #2:

The authors have addressed most of my concerns, and the language has been improved a lot. There are still some minor issues to be addressed:

1. Line 116-117, 229: *"The obtained chromosome-level genomes had a gap on Chr21 and Chr25,*

respectively. Subsequently, the gap on the Chr21 chromosome was successfully filled using TGS-GapCloser software and HiFi data". But in figure 1B, 50 chromosomes named Chr1A-12A, Chr1B-12B, and ChrB, which are not consistent with line 229 as well. Please confirm.

Response: We have carefully reviewed this and thank you for your correction. (Lines 115-117, Lines 234-238)

Lines 115-117: The obtained chromosome-level genomes had a gap on Chr 6B and Chr B, respectively. Subsequently, the gap on the Chr 6B chromosome was successfully filled using TGS-GapCloser software and HiFi data.

Lines 234-238: The subgenomes showed an overall 1:1 covariance but frequent chromosomal structural variations between some chromosomes, such as chromosomal inversions between Chr 8B and Chr 10A, Chr 12A and Chr 12B, and chromosomal translocations between Chr 10A and Chr 7B, and Chr 8A and Chr 10B (Figure S11).

2. Figure 1 needs to be modified, the Chromosomes are not legible.

Response: Thank you for your careful review, which is very important for our research. Please find attached a revised version of Figure 1, which has been redrawn for greater clarity.

3. The Chromosome names should be consistent. For example, in line 162, the chromosome labels have no space between the 'Chr' and the number (Chr7B, Chr7A, and Chr5B), but in line 229, there is a space between the 'Chr' and the number (Chr 9, Chr 12, Chr 19, and Chr 20).

Response: We have carefully reviewed this and thank you for your correction. (Lines 167-170)

Lines 167-170: We identified three miRNAs (Chr 7B, Chr 7A, and Chr 5B), two rRNAs (Chr 6B and Chr 4B), and two tRNAs (Chr B and Chr 11A) densely populated regions in the *P. australis* genome (Figure 1b j - m and Figure S8).

4. The language needs to be more concise.

Response: Thank you for your correction. Some of the descriptions in the manuscript have been edited for greater concision.

For example:

Lines 594-597: The rapid development of genome sequencing and assembly technologies is facilitating the assembly and understanding of an increasing number of complex genomes, particularly those of medicinal plants³⁷⁻³⁹. This has important implications for the identification of biosynthetic pathways for active compounds and genetic studies of adversity responses in these species.

Lines 539-545: The promoter regions of these SUC genes were found to contain multiple cis-acting regulatory elements associated with phytohormone (abscisic acid and MeJA) or stress response stress (defense and stress, light, and low temperature). Moreover, the binding sites of the transcription factor MYB were also identified, including MYB binding site involved in drought-inducibility, MYB binding site involved in light responsiveness, and MYBHv1 binding site. The findings indicate that stress-related MYB transcription factors may be a pivotal regulator of sucrose transporter proteins in *P. australis* (Figure 7b).

Lines 652-656: Subsequently, Bs retained genes favourable for functional maintenance and

structural stability through gene rearrangement and selection during long-term evolution. Concurrently, Bs preserved the stable configuration of the chromosome centromere and the transmission of genetic information through extensive methylation modifications.

Lines 748-751: The high expression of these genes promoted the accumulation of sucrose and polysaccharides in rhizomes and reduced the inhibition of rhizome axillary buds, facilitating the development of the complex rhizome network of *P. australis*.